# The dominant role of sunlight in degrading winter dissolved organic matter from a thermokarst lake in a subarctic peatland

Flora Mazoyer[1,2], Isabelle Laurion[1,2], Milla Rautio[2,3]

[1]Centre Eau Terre Environnement, Institut National de la Recherche Scientifique, Québec, QC, Canada
[2]Centre for Northern Studies, Université Laval, Québec, QC, Canada
[3]Département des sciences fondamentales, Université du Québec à Chicoutimi, QC, Canada

*Correspondence to*: Flora Mazoyer (mazoyer.flora@gmail.com)

**Abstract.** Dissolved organic matter (DOM) leaching from thawing permafrost may promote a positive feedback loop onto climate if it is efficiently mineralized into greenhouse gases. Yet, many uncertainties remain on the extent of this mineralization, which depends on DOM lability that is seemingly quite variable across landscapes. Thermokarst peatlands are organic-rich systems where some of the largest greenhouse gas (GHG) emission rates have been measured. At spring turnover, anoxic waters are releasing the GHG accumulated in winter, and the DOM pool is being exposed to sunlight. Here, we present an experiment where DOM photo- and bioreactivity were investigated in water collected from a thermokarst lake in a subarctic peatland during late winter (after six months of darkness). We applied treatments with or without light exposure, and manipulated the bacterial abundance with the aim to quantify the unique and combined effects of light and bacteria on DOM reactivity at ice-off in spring. We demonstrate that sunlight was clearly driving the transformation of the DOM pool, part of which went through a complete mineralization into $CO_2$. Up to 18% of initial dissolved organic carbon (DOC, a loss of 3.9 mgC $L^{-1}$) was lost over 18 days of sunlight exposure in a treatment where bacterial abundance was initially reduced by 95%. However, sunlight considerably stimulated bacterial growth when grazers were eliminated, leading to the recovery of the original bacterial abundance in about 8 days, which may have contributed to the DOC loss. Indeed, the highest DOC loss was observed for the treatment with the full bacterial community exposed to sunlight (5.0 mgC $L^{-1}$), indicating an indirect effect of light through the bacterial consumption of photoproducts. Dark incubations lead to very limited changes in DOC, regardless of the bacterial abundance and activity. The results also show that only half of the light-associated DOC losses were converted into $CO_2$, and we suggest that the rest potentially turned into particles through photo-flocculation. Sunlight should therefore play a major role in DOM processing, $CO_2$ production and carbon burial in peatland lakes during spring, likely lasting for the rest of the open-season in mixing surface layers.

## 1 Introduction

In northern regions, permafrost soil temperature is slowly increasing in response to climate change (Biskaborn et al., 2019), leading to permafrost thawing that has many consequences on biogeochemical cycles. Thawing affects the morphology of existing water bodies, and generates new ones through soil subsidence and erosion (Vonk et al., 2015b). When previously-frozen carbon stocks are mineralized into greenhouse gases, emissions could feed a positive feedback loop (Schuur et al., 2015), making the fate of this large pool of wide interest.

It is well established that freshwaters are a key component of the global carbon budget (Cole et al., 2007), and that carbon dioxide ($CO_2$) emissions from lakes are tightly linked to organic carbon inputs (Sobek et al., 2005). More recently, northern lakes were shown to be a significant part of the carbon cycle dynamic (Wik et al., 2016). The widespread thermokarst features created by abrupt thawing of ice-rich permafrost are particularly considered as hot spots of greenhouse gas emissions (e.g. Cory et al., 2013; Matveev et al., 2016). Olefeldt et al. (2016) estimated that peatland thermokarst landscapes cover 8% of the northern circumpolar permafrost area, and contain 15% of all organic carbon within the 0-3 m depth soil layer. Future changes in northern peatlands and the fate of these organic carbon stocks under warming thus deserve attention.

Mineralization of dissolved organic matter (DOM) heavily contributes to $CO_2$ emissions from inland waters (Lapierre et al., 2013; Tranvik et al., 2009), and mainly depends on two processes: bacterial respiration and sunlight oxidation (photooxidation). To estimate the respective shares of these phenomena or their importance at the scale of a lake or a region is a complex task, especially as they can influence each other (e.g. Obernosterer and Benner, 2004) and are constantly changing over space and time (reviewed by Cory and Kling, 2018). Seasonal variations in $CO_2$ emission rates have been studied in various types of northern lentic ecosystems (Elder et al., 2018; Hughes-Allen et al., 2021; Prėskienis et al., 2021; Sepulveda-Jauregui et al., 2015), underlining the importance of collecting data on a complete annual cycle. Because the summer season offers easier conditions for sampling and instrument handling, most studies of northern lakes have occurred during this period (Block et al., 2019). The long ice-covered winter and the short ice-breaking spring seasons have historically been overlooked by limnologists, although many northern lakes are covered by ice for several months (Hampton et al., 2017). In fact, recent studies demonstrate that carbon dynamics during winter and at ice-break are far from negligible. The review by Denfeld et al. (2018), presenting results on boreal lakes from Scandinavia and North America, evaluated that 17% of annual $CO_2$ emissions happened at the ice-break period, while the study from Jansen et al. (2019) reports up to 30%. These emissions would mainly result from a fast release of gases accumulated under the ice over the winter season, and from DOM photomineralization that is particularly intense at spring (Vachon et al., 2016, 2017). In northern lakes, sunlight was responsible for up to 49% of DOM mineralisation at spring time, in comparison to only 14% during the open water season (Vachon et al., 2016). The concept of DOM photoreactivity appears to be particularly central to explain these seasonal or temporal differences (Groeneveld et al., 2016; Pickard et al., 2017).

The goal of our study was to experimentally explore the photoreactivity and bioreactivity of late-winter DOM collected from a thermokarstic lake located in a subarctic peatland of eastern Canada. This work is a contribution on learning about carbon cycling in lakes particularly rich in organic matter during the critical ice-off transition period, when DOM microbially processed during the winter becomes exposed to sunlight, higher temperature and oxygenation in surface layers. Winter DOM reactivity has been identified as a major knowledge gap in carbon biogeochemistry research (Hampton et al., 2017), and thermokarst lakes in non-yedoma areas are overlooked systems (Vonk et al., 2015b). The objectives of this study were to assess the effects of sunlight on DOM properties, microbial dynamics and carbon biogeochemistry, as compared to dark processing. We hypothesized that sunlight would have a dominant effect on DOM degradation and mineralization after the long subarctic winter, and a positive effect on bacterial metabolism through the phototransformation of aromatic compounds. We tested this through outdoor incubations where DOM properties, $CO_2$ concentration, bacterial abundance and bacterial production were followed in replicated microcosms over 18 days.

**2 Material and methods**

 **2.1 Study site**

The study site was located on the eastern shore of Hudson Bay, 8 km southwest of the village of Whapmagoostui-Kuujjuarapik, Nunavik (Quebec, Canada), in the peatland valley of the Sasapimakwananisikw River, hereafter referred as the SAS valley. The historical development of this subarctic palsa site has been reconstructed through palaeoecology in Arlen-Pouliot and Bhiry (2005). Peat accumulation over the marine clay bed started shortly after

 6000 cal. BP and interrupted 400 cal. BP with the Little Ice Age. During this cooling period, permafrost established on this ombrotrophic bog and caused the uplifting of palsas (round-shaped mounds). Since about 100 years, temperature and precipitation have increased again, leading to permafrost thaw, palsa collapse, and thermokarst lake formation. These black-coloured water bodies lay in a peat bog mainly colonized by ericaceous shrubs, semi-aquatic plants such as *Carex aquatilis*, *Sphagnum* sp. and brown mosses (Bhiry et al., 2011). Nowadays, permafrost

 in the SAS valley underlies less than 2% of the land surface, and gathers in the core of the scattered palsa mounds (Bhiry et al., 2011). Detailed landscape diagram and picture can be found in Figure 7 of Vincent et al. (2017).

**2.2 Field sampling and in situ measurements**

Sampling was carried out between the 19 and 24 March 2016 in a thermokarst lake named SAS2A, with an area of 196 m² and a maximum depth of 2.8 m (55.225018°N, 77.696580°W). It took place approximately two months

 and half before the complete disappearance of the ice cover (30th May). More information about this well-studied lake, along with a description of the winter sampling campaign, are further reported in Matveev et al. (2019). The lake was covered with 0.5 m of snow and 0.6 m of ice, underlain with a water column of about 1.7 m, which was entirely anoxic. At this time of the year, temperature varied between 0.5°C (immediately below the ice) and 1.9°C (bottom). Despite this thermal stratification, DOM properties were similar through the water column: DOC varied

 between 18.3 and 20.9 mgC $L^{-1}$, SUVA between 6.78 and 6.85 L mgC$^{-1}$ m$^{-1}$, and $S_{285}$ between 0.0111 and 0.0105 nm$^{-1}$ (Table S1). Water for the experiment was collected on 24 March at the surface just below the ice cover, and stored in a 10-L carboy. In situ data demonstrated that this water was well representing the whole water column in terms of DOM properties and nutrients (Table S1).

**2.3 Experimental setup**

 In this paper we use the term *bacteria* for simplicity, to refer to any heterotrophic planktonic microorganisms affecting the DOM pool with a maximum size of about 3 µm, which may include archaea, phototrophic eukaryotes and small protozoans. On the other hand, cell counts obtained by flow cytometry only comprise cells smaller than about 1 µm (see section on bacterial abundance below). Also, the term *light* used hereafter refers to sunlight wavelengths involved in photooxidation, thus it includes ultraviolet radiation and not only visible radiation.

 For logistical reasons, the experiment was set up approximately four months after the water was collected in the field, on the roof of the INRS in Quebec City (46.812899°N, 71.223821°W in Quebec, Canada). Some DOM changes were noticed between water collection and the beginning of the experiment (see Table S1 to compare with Table 1), where we can observe increases in dissolved organic carbon (DOC, by 15%) and in fluorescing DOM (by 64%), along with decreases in chromophoric DOM (by 17% at 320 nm) and in aromaticity (by 28%). We

assumed these changes to be representative from what is naturally happening in the lake over the long winter as organic matter leaches from particles and DOM and bacteria interact under the ice. Oxygenation at water collection was likely limited as it was done with a thin-layer sampler connected to a peristaltic pump, and SAS2A water was shown to be very resistant to oxygenation (Folhas et al., 2020). The collected water was thus kept under similar conditions as found under the ice (dark, 4°C, no oxygenation since the container was filled to the top in the field). These physico-chemical conditions are structuring this type of ecosystem in winter, where there are no inputs of fresh DOM that could drive bacterial community or DOM pool changes (Bertilsson et al., 2013). Ultimately, we considered the experimental water to contain DOM and bacterial community representative of the winter, even after a 4-months delay.

**Table 1. Water characteristics at the start of the incubation ($T_0$), before applying the different treatments: BL = with bacteria and light, B = with bacteria in the dark, L = bacteria-filtered with light, C = bacteria-filtered in the dark and PI = with bacterial inoculum in the dark after a light pre-incubation. Characteristics include dissolved organic carbon (DOC), dissolved inorganic carbon (DIC), absorption coefficient of DOM at 320 nm ($a_{320}$), specific absorbance index at 254 nm ($SUVA_{254}$), absorption slope at 285 nm ($S_{285}$), total fluorescence by DOM ($F_{tot}$ as the sum of the 5 components) and its proportion into C1-C5 components, total bacterial abundance (BA), and the bacteria production rate (BP). Variables are given as the triplicate mean ± standard error.**

|  | Water with the original bacterial community* | Bacteria-filtered water† | Pre-incubated water inoculated‡ |
|---|---|---|---|
| Associated treatments | BL, B | L, C | PI |
| DOC ($mgC\ L^{-1}$) | $22.1 \pm 0.7$ | $21.9 \pm 0.7$ | $19.8 \pm 0.2$ |
| DIC (mM) | $0.32 \pm 0.01$ | $0.25 \pm 0.01$ | $0.29 \pm 0.01$ |
| $a_{320}$ ($m^{-1}$) | $119 \pm 0.2$ | $115 \pm 0.5$ | $99 \pm 0.9$ |
| $SUVA_{254}$ ($L\ mgC^{-1}\ m^{-1}$) | $4.68 \pm 0.14$ | $4.56 \pm 0.13$ | $4.67 \pm 0.3$ |
| $S_{285}$ ($nm^{-1}$) | $0.0100 \pm 10^{-5}$ | $0.0100 \pm 10^{-5}$ | $0.0110 \pm 10^{-5}$ |
| $F_{tot}$ (RU) | $8.2 \pm 0.01$ | $8.2 \pm 0.05$ | $6.5 \pm 0.02$ |
| C1 (RU) / % of $F_{tot}$ | 2.8 / 34 | 2.8 / 34 | 2.3 / 35 |
| C2 (RU) / % of $F_{tot}$ | 2.0 / 24 | 2.0 / 24 | 1.8 / 28 |
| C3 (RU) / % of $F_{tot}$ | 1.6 / 19 | 1.6 / 19 | 0.9 / 14 |
| C4 (RU) / % of $F_{tot}$ | 1.2 / 15 | 1.2 / 15 | 1.2 / 18 |
| C5 (RU) / % of $F_{tot}$ | 0.7 / 9 | 0.7 / 9 | 0.4 / 6 |
| BA ($10^5\ cells\ mL^{-1}$) | $13.02 \pm 2.26$ | $0.69 \pm 0.18$ | $1.95 \pm 0.26$ |
| BP ($\mu gC\ L^{-1}\ h^{-1}$) | $0.224 \pm 0.017$ | $0.003 \pm 0.001$ | $0.015 \pm 0.001$ |

*Filtration with nominal 1.5 µm glass fiber filters to remove most grazers. †Filtration with 0.2 µm filters to remove most bacteria (95%) and all grazers; ‡Pre-filtration with 0.2 µm filters for a 2-days incubation under sunlight, followed by the inoculation with 3-µm filtered water at a volumetric ratio of 10% and incubated in the dark.

The preparation of the experiment started with series of filtration steps to generate two types of water, one containing the original bacterial community (1.5-µm filtered) and the other with a largely reduced bacterial abundance (0.2-µm filtered). While we tried to achieve a sterile sample, flow cytometry results later indicated that the original abundance was only reduced by 95%. The filtration steps included pre-filtration through 64 and 5 µm sieves using Milli-Q rinsed NITEX®, followed by filtration through pre-combusted 1.5-µm glass fibre filters (VWR®, nominal porosity) to remove heterotrophic nanoflagellates and non-living particles. The study from del Giorgio and Bouvier (2002) reported that 1-µm glass fibre filters show good results for that purpose. We observed that filters were brittle during the filtration steps (potentially related to the pre-burning step), and we suspect that cracks may have existed. Part of this water was kept for the incubations with bacteria, and the rest was

subsequently passed through a sterile 0.2-µm capsule filter (HT Tuffryn® polysulfone membrane, Pall Corp.), or after the faster-than-expected capsule clogging, a double filtration step using pre-combusted 0.7-µm glass fibre filters (GF75, nominal porosity, Advantec®) and pre-rinsed 0.2-µm cellulose acetate filters (Advantec®) to remove bacteria. Water filtered by these two methods was merged to obtain a homogeneous 0.2-µm filtrate.

These two types of waters were exposed to natural sunlight or dark conditions on the roof of the laboratory during 18 days, from 30 July to 17 August 2016. Five treatments were aimed from various combinations of "bacteria" and "light". In four of them, these factors were applied simultaneously: with bacteria and light (hereafter BL), with bacteria in the dark (B), bacteria-filtered with light (L), and bacteria-filtered in the dark (C, for control). The fifth treatment was applied in a consecutive manner: a portion of the bacteria-filtered water was pre-incubated under

natural sunlight for two days (approximately 42.6 MJ m$^{-2}$ received in total), and then inoculated with bacteria for an incubation in the dark (PI, for pre-incubated). The goal of this last treatment was to uncouple the effect of sunlight from the microbial degradation processes. The bacterial inoculation was done using original lake water filtrated through 3-µm polycarbonate filters (PCTE membrane, Advantec®), and using a volumetric ratio of 10%. The choice of using 3-µm filters (made for logistical reasons) was considered fairly close to using nominal 1.5-

µm filters as for B and BL treatments. Overall, the preparation of the experiment lasted about a week, during which water was systematically kept in the dark at 4°C.

Water incubation was carried out in 60 mL transparent Teflon™ FEP bottles (Nalgene™), a material known for its transparency to ultraviolet and visible radiation (above 90% as shown in Fig. S1 of Logozzo et al., 2021) at the same time of being less fragile to handle than quartz. Twelve bottles per treatment were filled to the rim and

incubated, allowing to sacrifice triplicate bottles at days 3, 8, 13 and 18. Dark treatment bottles were wrapped in opaque tape. Bottles were all immersed at the same time in a tray placed on the roof top. The tray was equipped at the bottom with a twisted copper pipe connected to a circulating bath aiming at limiting large increase of temperature during sunny days and tempering day-night differences. The tray was filled with tap water (added daily to offset evaporative losses), and oriented to maximize sunlight exposure, with bottles moved daily to limit

unequal shading. Three light and temperature loggers (HOBO Pendant®, ONSET) were installed at the same depth as the bottles to obtain a relative index of incoming sunlight, with one wrapped as in the dark treatment to control for temperature difference among treatments. Temperature in the light averaged 19.7°C (ranging from 8.5 to 34.3°C) as compared to 18.9°C (8.4 to 32.0°C) in the dark (Fig. S1 a). These temperatures were higher than the ones measured in SAS2A at the spring ice-break, but averages were similar to temperatures measured at the pond

surface in summer (Matveev et al., 2019).

Time zero (T$_0$ – at day 0) measurements were taken from the three types of water: water containing the original bacterial community (BL and B), the bacteria-filtered water (L and C) and the inoculated pre-incubated water (PI). They were the only pseudo-replicates from the dataset (i.e., 3 samples taken from the same container, as opposed to 3 replicate bottles for days 3, 8, 13 and 18). At each time step, we measured DOC, dissolved inorganic carbon

(DIC), chromophoric dissolved organic matter (CDOM, absorption spectra), fluorescing dissolved organic matter (FDOM, excitation-emission matrices) and bacterial abundance. Bacterial production was only measured at T$_0$, day 8 and day 18. Note that filtration steps allowed to oxygenate the water, so that all treatments at T$_0$ had an overall mean of $8.96 \pm 0.23$ mgO$_2$ L$^{-1}$.

**2.4 Incoming radiation during the experiment**

Radiation during the incubation was measured every 2 minutes in a weather station (Wireless Vantage Pro2™ Plus, Davis Instrument Corporation) installed on Laval University campus (http://meteo-laval.gel.ulaval.ca/meteocam-proxy.php) at a distance of 5.4 km from INRS (pyranometer integration from 300 to 1100 nm). Missing data during the first three days were estimated through a linear regression between the station irradiance and our Pendant logger values. Hourly radiation measured at the SILA station of Whapmagoostui-

Kuujjuarapik in 2015 and 2016 (on the shore of Hudson Bay, 8 km away from the study site, pyranometer integration from 300 to 2800 nm; CEN, 2020a) and at the SAS study site from 2017 to 2019 (local radiation unavailable before 2017, same pyranometer; CEN, 2020b) were integrated over the incubation period (18 days in August). They were compared to get a sense of incoming sunlight energy at both latitudes (Fig. S1 b). Total incoming sunlight energy was 33 to 103% higher in Quebec City (349 MJ m$^{-2}$) than at SAS site (from 172 to 263

MJ m$^{-2}$), but similar to the sunlight energy recorded at the SILA station (309 and 342 MJ m$^{-2}$).

**2.5 DIC analysis**

The production of dissolved $CO_2$, as a measure of bacterial respiration and DOM photomineralization, was estimated through changes in DIC concentration, assuming that any increase in DIC over time within the incubation bottles corresponded to a production of $CO_2$. Exetainer® vials (12 mL, Labco) were prepared

beforehand by adding 120 µL of 10% HCl, and manually purging with gaseous nitrogen for one minute. Quickly after opening the incubation bottle, 3 mL of water was gently sampled with a syringe and injected with a 21G needle through the septum of a prepared vial. The vial headspace was analysed by gas chromatography within two months (TRACE™ 1310 GC Greenhouse, Thermo Fisher Scientific, Molecular Sieve column 5A 80/100 6' x 1/16'', thermal conductivity and flame ionisation detectors, calibration curve made with carbonate solutions 0-2.5

mM). Vials were weighted at each step to convert masses to volumes, and DIC was calculated as following Eq. (1):

$$[DIC] = \frac{[CO_2(g)] \times (V_{vial} - V_{gas} + 1.2 \times V_{gas})}{1.2 \times V_{sample}}, \tag{1}$$

where [DIC] is DIC concentration in mM, [$CO_2$(g)] the concentration of $CO_2$ contained into the gaseous phase in mM, $V_{vial}$ the vial volume in mL, $V_{gas}$ the gas volume contained in the vial in mL, and $V_{sample}$ the sample volume

added to the vial in mL. The value of 1.2 is the Henry's law volatility constant $K_H$ for $CO_2$.

**2.6 DOM analyses**

Water samples were filtered onto 0.45-µm polyethersulfone syringe filters (AMD Manufacturing Inc.) and stored in 40-mL glass vials at 4°C. Part of this water was used to perform the optical analyses within 24h after sampling, and the rest was stored for later DOC analysis (within 2 months) on a Total Organic Carbon analyser (Aurora

1030W, O.I. Analytical) using the persulfate oxidation method (precision < 5%, detection limit 0.1 mg L$^{-1}$). CDOM absorbance scans between 200 and 800 nm on samples brought at room temperature were obtained with a dual beam spectrophotometer (2 nm slits, scan speed 200 nm min$^{-1}$, Cary 100 Bio, Varian) using a 1-cm path quartz cuvette. The null-point adjustment was applied on blank-corrected absorbance spectra using the mean value from 790 to 800 nm, and data were converted to Napierian absorption coefficients (in m$^{-1}$). The absorption

coefficient at 320 nm (hereafter, $a_{320}$) is used as a proxy for CDOM quantification. The specific absorbance at 254 nm (SUVA$_{254}$), calculated following Weishaar et al. (2003), is used as a proxy for the aromatic content of DOM. The spectral slope between 275 and 295 nm ($S_{285}$, in nm$^{-1}$), as a linear fit on log-transformed data, was also calculated, commonly used as an indicator of the mean size of DOM molecules (Helms et al., 2008).

Fluorescence excitation-emission matrices (EEMs) were obtained on a spectrofluorometer (Cary Eclipse, Varian) across an excitation waveband from 240 to 450 nm (10 nm increment), and an emission waveband from 300 to 560 nm (2 nm increment). DOM was characterized by the extraction of fluorescent components using the parallel factor analysis (PARAFAC) developed by Stedmon et al. (2003). Blank-subtracted EEMs were corrected for inner-filter effect using absorbance spectra and standardized to Raman units (RU), after which the Raman and Rayleigh scatter peaks were excised. All pre-processing was done with the FDOMcorr toolbox for Matlab (version 1.4, Murphy et al., 2010), and the model was created using the drEEM toolbox (version 0.1.0, Murphy et al., 2013). Extra samples from another experiment done on the same pond (N=54; unpublished data) were added to the 60 samples of the current experiment in order to obtain a minimum of 100 matrices to generate the PARAFAC model (according to Stedmon and Bro, 2008). A 5-component model was validated using split-half analysis, and the residuals did not present any obvious peak signals.

Fluorescence components C1 to C5 were expressed as the maximum fluorescence intensity at the peak in RU. The total fluorescence ($F_{tot}$) was obtained by summing the fluorescence of all 5 components, and used to calculate their relative abundance. The identification of the components was done in the Openfluor database of Murphy et al. (2014) through the selection of similar spectra with a Tucker congruence coefficient (TCC) exceeding 0.95 on the excitation and emission spectra simultaneously. Unfortunately, the EEMs from $T_0$ could not be exploited (related to an acquisition problem). However, based on CDOM dynamics from $T_0$ and FDOM dynamics from T3 (see results), we reasonably estimated the initial values of treatments B, C and PI to be the same as at T3, and the initial values of BL and L to be similar to the ones of B and C, respectively. These values were therefore used to calculate temporal changes and make statistics hereafter.

### 2.7 Bacterial abundance and production

Bacterial abundance (BA) was obtained on water samples fixed with glutaraldehyde (1% final concentration), snap-frozen with liquid nitrogen, and kept at -80°C until analysed with a flow cytometer (Accuri C6 Plus, BD Biosciences). Before counting, the samples were unfrozen at 4°C overnight and then brought to room temperature. Cells were stained with SYBR Green I (2.5X final concentration) for 15 minutes in the dark. Data were acquired at the slowest flow rate, with speed measured daily using Trucount$^{TM}$ absolute counting tubes (mean of 0.43 µL sec$^{-1}$ over the analytical period). Yellow-green Fluoresbrite® microspheres of 1 µm diameter (Polysciences Inc.) were added directly to Trucount tubes as an internal size standard and to control gate positions. Cells were discriminated on a log-scale using green fluorescence and forward-scatter for a blue laser excitation. One acquisition was done for each sample at a maximum rate of 1000 events sec$^{-1}$ and with a minimum of 10 000 events counted. Gating was refined considering all samples in order to extract their population abundance using the same gates (Fig. S4). The presence of picophytoplankton cells was checked in samples exposed to sunlight using cytometer runs done without staining with SYBR Green and looking at fluorescence from chlorophyll-a or phycocyanin, but no population could be discriminated from the noise. Samples were analysed before and after

sonification (Sonifier® SFX150, Branson™, 30 sec. with pulses of 0.1 sec./sec. at 50% duty cycle, ~ equivalent to 10 W) to evaluate the relative proportion of particle-attached and free-living bacteria. The sonication rod was directly introduced into the acquisition tubes placed in an ice bath.

Bacterial production (BP) was measured using the tritiated leucine incorporation method (Kirchman et al., 1985), with centrifugation improvement proposed by Smith and Azam (1992). Production assays started less than 3 hours after sampling. A working solution with a specific activity of 150 Ci mmol$^{-1}$ was added to triplicate aliquots, each containing 1.2 mL of sample water (final leucine concentration of 30 nM). Incubations for 2 hours in the dark at room temperature (~21°C) were ended with the addition of trichloroacetic acid (TCA, 5% final concentration). For each triplicate, one additional tube was fixed with TCA at the beginning of the incubation to account for passive incorporation of tritiated leucine by cells. The tracer concentration and incubation duration were based on previous studies on thermokarst lakes near the sampling site (Roiha et al., 2015). Tubes were then stored at -20°C until analysis. Samples were melted at room temperature, centrifuged 10 minutes at 16 000 g, and the pellet rinsed 2 times with 5% TCA solution. Scintillation cocktail (Ecolite(+)$^{TM}$, MP Biomedicals) was added for at least 24 hours before counting with a liquid scintillation analyser (Tri-Carb® 2910TR, Perkin Elmer). Counts per minute were corrected with controls and converted into disintegration per minute (DPM) using the counting efficiency calculated from home-made standards. Sample aliquots acting as controls were prepared with the same steps described above but without adding the tracer solution, while standards were aliquots with a known amount of radioactive leucine. Calculation details on BP can be found in the supplementary information.

**2.8 Data analysis**

The experimental design was analysed with ANOVAs excluding $T_0$ values. Effects and interactions among BL, B, L and C treatments were examined with a 3-way ANOVA testing for light (2 levels, fixed), bacteria (2 levels, fixed) and time (4 levels, fixed) on each variable. PI treatment was not included in this analysis due to its distinct preparation method, but it was compared to the other treatments separately using a 2-way ANOVA testing for treatments (5 levels, fixed) and time (4 levels, fixed). ANOVA assumptions were checked by graphical examination of the residuals plotted against the expected values. Except for C5, all variables were transformed to validate assumptions of the ANOVA. A log-transformation was applied to $S_{285}$, while all remaining variables were transformed with square root. When a term of the ANOVA was significant, comparison between the different levels was obtained using Tukey's Honestly Significant Difference posthoc test. Wilcoxon test was occasionally used to test for two-groups mean differences. A significance level of $\alpha = 0.05$ was used for all statistical analysis. The 95% confidence interval around $T_0$ values were calculated and represented in the figures as a thin bar on the left side of $T_0$. Values falling outside this interval during the incubation were considered as significantly different from $T_0$. ANOVAs were performed with JMP 15.0 software, while the rest of the statistical analyses and graphs were done on RStudio 1.2.5 software.

**3 Results**

### 3.1 Initial characteristics of incubated waters

At $T_0$, DOM properties between waters with or without bacteria were similar, showing that the filtration step did not affect DOM composition (Table 1). As expected, bacterial abundance and production were much higher in the water containing the initial bacterial community, with 19 times more free bacterial cells, and 75 times higher production rate (BP normalized by BA was also 4 times higher on average). Despite our sterilization efforts relying on 0.2-µm filtration, approximately 70 000 cells per mL remained in the bacteria-filtered water (i.e. 5% of the original abundance). This implies that L and C treatments were not completely sterile, but this filtration step apparently removed all protists (see below).

The pre-incubated water (PI treatment) displayed different values for all variables except $SUVA_{254}$ as compared to the two other $T_0$ water types, which was expected after an exposition to sunlight for 3 days. DOC, $a_{320}$ and $F_{tot}$ were lower than for the two other waters, while $S_{285}$ was higher (Table 1). There had been an increase of DIC during the pre-treatment with light (by approx. 16%) when comparing values with those in bacteria-filtered water from which PI treatment water was prepared (this cannot be seen in Fig. 1 since DIC at $T_0$ was subtracted from all values). Bacterial abundance and production were also higher in PI than in C treatment, showing the effect of the bacterial inoculation, but remained considerably lower than in the water containing the original bacterial community. The five fluorescent components were less abundant in the pre-incubated water than in the two other waters. Overall, C1 and C2 were the most abundant fluorophore groups (~60% in total), C5 was the least abundant (<10%), and C3 and C4 were approximately present in the same proportions (~15% each).

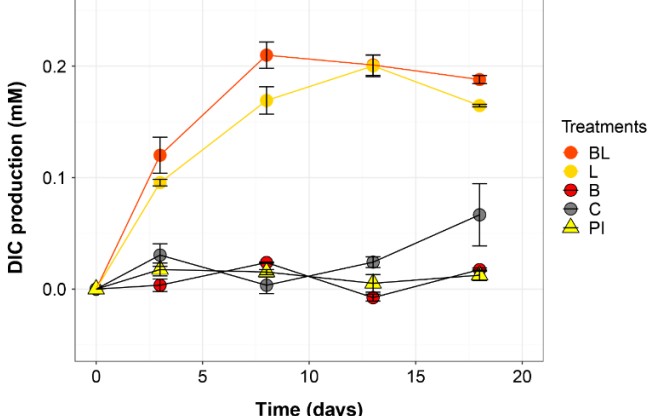

**Figure 1. Temporal dynamics in dissolved inorganic carbon (DIC), presented as the difference from the mean value at time zero for each treatment (i.e. the DIC produced during the incubation). Error bars represent the standard errors. Treatments are as follows: BL = water with original bacterial community exposed to light, L = bacteria-filtered water exposed to light, B = water with original bacteria incubated in the dark, C = bacteria-filtered water incubated in the dark, and PI = pre-incubated water inoculated with bacteria and incubated in the dark.**

All five DOM components were identified as humic-like, with maximum emission peak above 400 nm (see all supporting references in Table S2, and Fig. S2). C1 is broadly described as a common ubiquitous group of fluorophores with a terrestrial origin (e.g. Williams et al., 2010). C2 was most probably of terrestrial origin (found dominant in wetland and forest drainage waters by Søndergaard et al., 2003) and described as bio-refractory. Component C3 showed the longest and broadest excitation band, along with the longest emission peak, suggesting that it belongs to a particularly high-molecular weight fraction of terrestrial humic matter (Stedmon et al., 2003). It was reported in very diverse environments and usually assigned to a terrestrial origin (e.g. Osburn et al., 2012),

but some studies assigned it to an autochthonous origin. Note that Murphy et al. (2018) demonstrated that humic-like components from terrestrial and autochthonous sources could display similar spectra and chemical characteristics, thus highlighting the uncertainty related to the origin of fluorescing components. Finally, even though the origin of C4 and C5 is uncertain, they could be characterized by their photodegradability.

**3.2 Changes in chemical and optical variables during the incubation**

The largest changes in DIC, DOC and various CDOM properties were observed for the light treatments BL and L (complete ANOVA results presented in Table S3). DIC (Fig. 1) significantly accumulated in bottles in both light treatments until day 8 (posthoc, $P < 0.0001$), at respective rates of 0.025 and 0.020 mM day$^{-1}$, after which it remained stable (posthoc, $P > 0.97$). Correspondingly, concentrations in DOC and CDOM (Fig. 2, with $a_{320}$ as a proxy for CDOM concentration) significantly decreased with light, with total losses of 6.7 mgC L$^{-1}$ (by 30%) and 47 m$^{-1}$ (40%) for BL, and 5.6 mgC L$^{-1}$ (25%) and 45 m$^{-1}$ (39%) for L after 18 days of incubation.

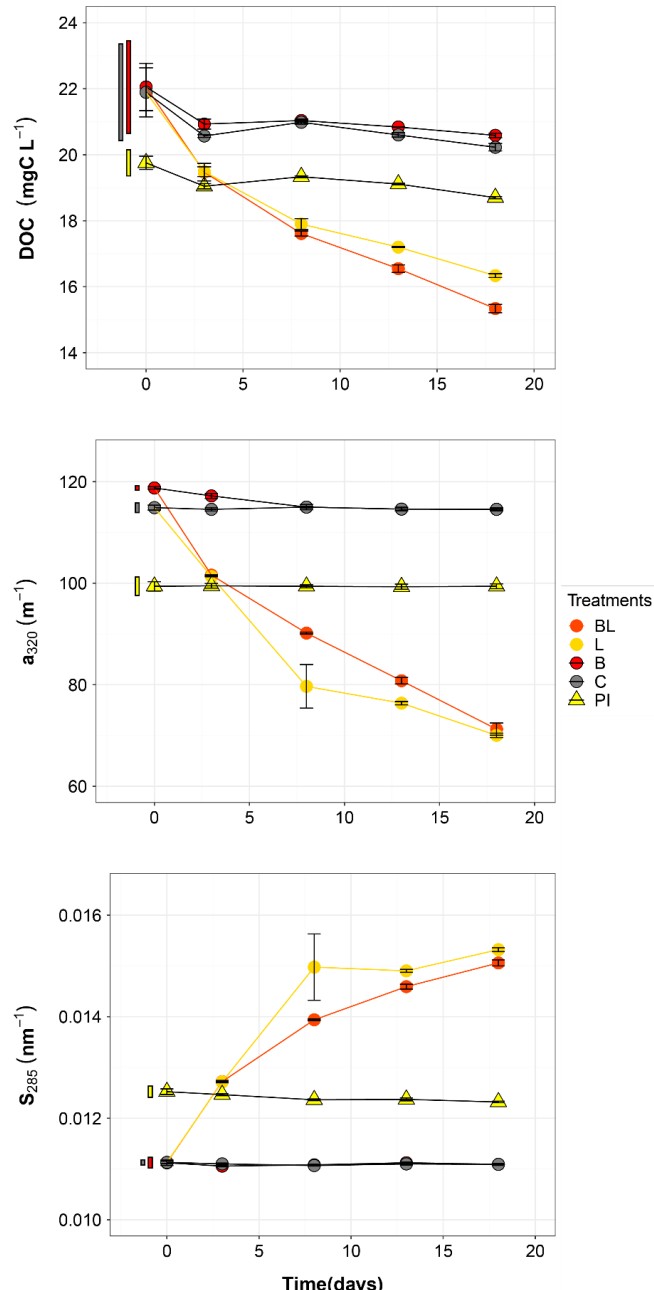

**Figure 2. Temporal dynamics of the dissolved organic carbon (DOC), the absorption coefficient at 320 nm ($a_{320}$) and the absorption slope at 285 nm ($S_{285}$) for the five treatments (given in Fig. 1). Error bars represent the standard errors. The thin bars on the left of $T_0$ values indicate the 95% confidence interval at $T_0$ using the corresponding colours of the three water types of Table 1 (grey = 0.2 µm-filtered; red = 1.5 µm-filtered; yellow = 0.2 µm-filtered, pre-incubated for 3 days and inoculated with bacteria). Treatment abbreviations are provided in caption of Figure 1.**

DIC, CDOM and DOC values were not statistically different between the two light treatments, with the exception of CDOM at day 8 (posthoc, P < 0.0001), and DOC at days 13 and 18 (posthoc, P < 0.0126). Significant increases in $S_{285}$ (Fig. 2) were also observed in the light treatments, indicating a decline in DOM overall molecular size over time. There was a significant decreasing trend in $SUVA_{254}$ during the experiment, but values did not get out of the 95% confidence interval of $T_0$ (data not shown). As for CDOM, there was an overall decrease in FDOM under light (Fig. 3).

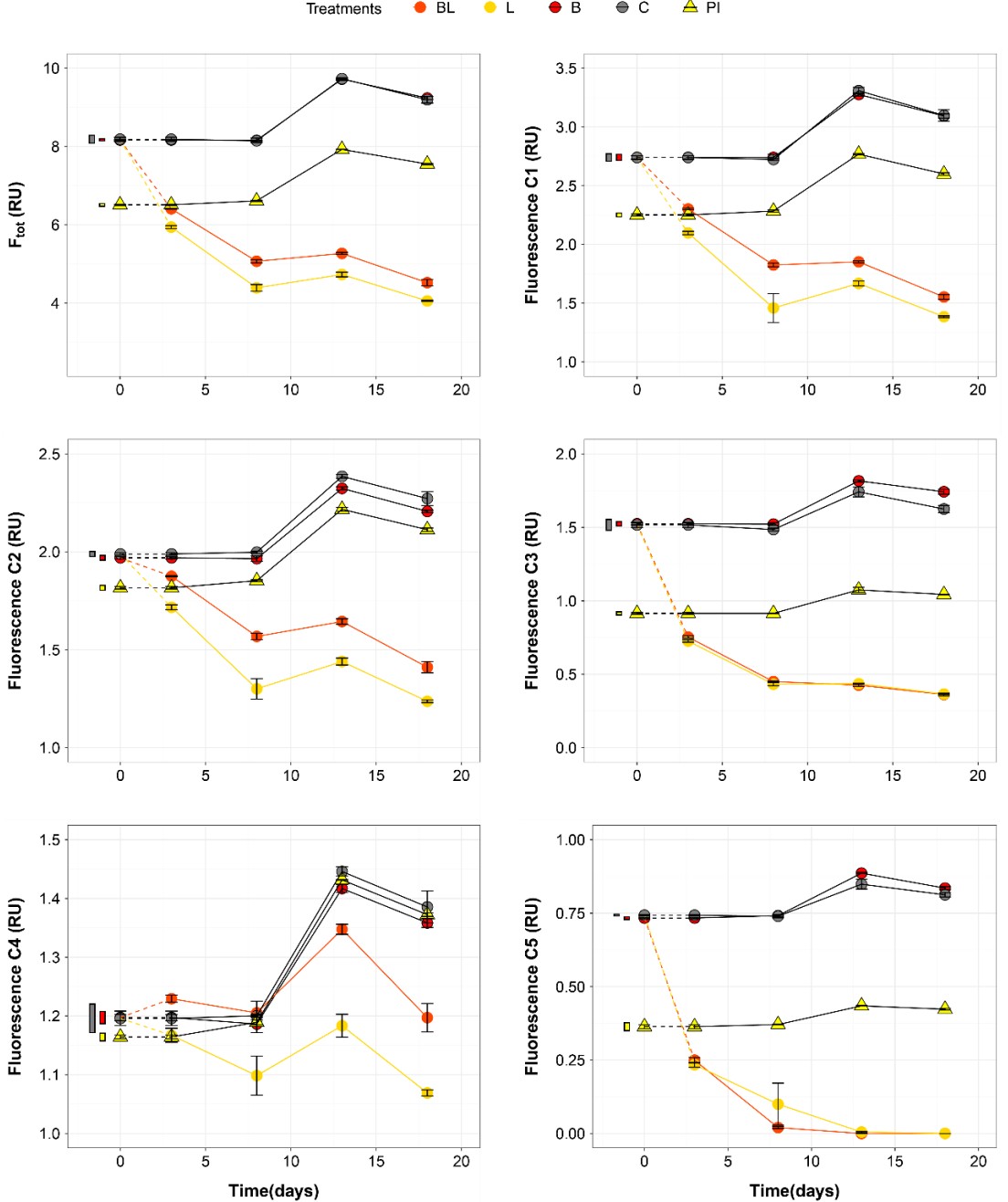

**Figure 3. Temporal dynamics of the total fluorescence ($F_{tot}$) and the five fluorescing components (C1-C5) over the 18 days of incubation for the five treatments (given in Fig. 1). Error bars represent the standard errors. The thin bars on the left of $T_0$ values indicate the 95% confidence interval at $T_0$ (colour codes as in Fig. 2 caption). Treatment abbreviations are provided in caption of Figure 1.**

After 18 days, $F_{tot}$ losses were 3.6 RU (by 45%) and 4.1 RU (50%) for BL and L, respectively. C1 and C3 contributed the most to these losses (around 30% each), followed by C2 and C5. C5 completely disappeared after 13 days of light exposure. On the other hand, C4 did not vary much but still showed a significant decrease under light exposure. The decreasing dynamic strongly slowed down starting from day 8 for almost all variables.

On the contrary to the dynamics observed under sunlight exposure, there were almost no changes over time for the dark treatments (B and C). Only at day 18, DIC was found significantly higher in C treatment by comparison to

the rest of the incubation, and the DOC value was out of the 95% confidence interval of $T_0$ (Figs. 1 and 2). Total DOC losses were 1.5 and 1.7 mgC L$^{-1}$ for B and C treatments, respectively. A sudden increase in all fluorescent components was observed after day 8 (mean increase by 19%) (Fig. 3). There was no overall difference in the

measured variables for B and C treatments, except for DIC, DOC and C3 fluorescence.

The variables measured in the PI treatment showed responses that were comparable to those of the dark treatments, with limited or no changes in time (Table S4). FDOM components, except C5, significantly increased after 8 days (posthoc, $P < 0.0004$) in the same proportions as in treatments B and C (mean increase by 19%). Over 18 days, there was a DOC loss of 1.1 mgC L$^{-1}$ that was not reflected in CDOM (loss <0.01 m$^{-1}$). As expected, initial DOM

properties in PI treatment such as DOC, $a_{320}$, $S_{285}$ or C1 were close to day 3-values of L treatment since the pre-incubation with light lasted about 3 days (posthoc, $P > 0.30$ at day 3).

### 3.3 Bacterial abundance (BA) and production (BP)

The BA at $T_0$ already indicated that sterility was not achieved by the 0.2 µm filtration step (Table 1), and this initial population significantly increased in abundance over time in both L and C treatments (respectively 39 and

4 times more at day 18 compared to $T_0$; Fig. 4). The most outstanding dynamic was obtained for the L treatment, with an overall growth of 1.5 x 10$^5$ cells mL$^{-1}$ day$^{-1}$. This means it took about 8 days for BA in the 0.2 µm-filtered L treatment to reach the initial abundance observed in B and BL treatments, and by the end of the experiment, BA had reached 2.69 x 10$^6$ cells mL$^{-1}$. The bacterial population that bypassed the 0.2-µm filtration, or was airborne during filling the bottles, grew at a much slower pace in the control treatment (0.1 x 10$^5$ cells mL$^{-1}$ day$^{-1}$), and

declined after day 13. The BA was stable in treatments with the original bacterial community (BL and B), with values still inside the 95% confidence interval of $T_0$ at day 18. However, bacteria remained overall more abundant in BL than in B treatment (posthoc, $P < 0.0001$). Growth in PI treatment was significant but BA remained stable between day 3 and day 18. At the end of the incubation, there were more bacteria in PI than in C (posthoc, $P = 0.04$).

Bacterial production significantly increased from $T_0$ in all treatments during the incubation (Fig. 5). The highest production rates were obtained for the BL treatment, reaching 1.11 µgC L$^{-1}$ h$^{-1}$ by the end of the incubation, equivalent to twice the production in the dark (B treatment) at the same moment. The exposition to sunlight and the presence of the initial community both contributed to the higher BP (Table S3). However, for treatments with a large reduction of the original bacterial abundance (L and C), the BP at day 18 was lower than at day 8 (posthoc,

$P = 0.04$), which is particularly evident for the L treatment. For PI treatment, BP did not change significantly between day 8 and day 18 (posthoc, $P = 0.9957$).

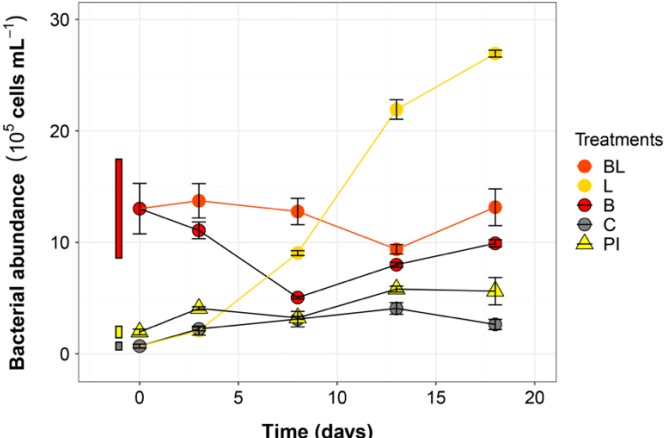

**Figure 4. Bacterial abundance along the 18-day incubation for the five treatments (given in Fig. 1). Error bars represent the standard errors. The thin bars on the left of $T_0$ values indicate the 95% confidence interval at $T_0$ (colour codes as in Fig. 2 caption). Treatment abbreviations are provided in caption of Figure 1.**

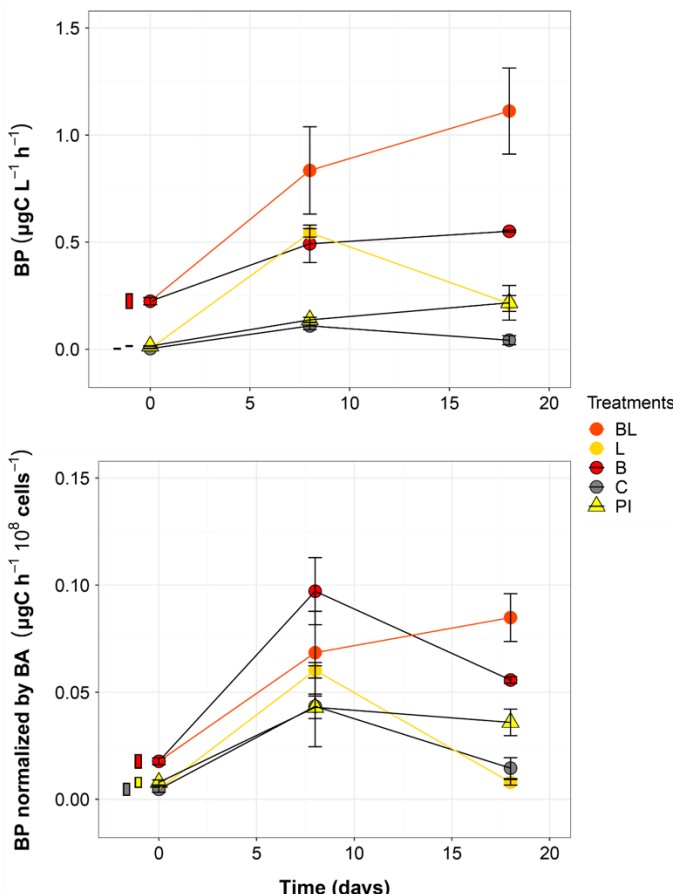

**Figure 5. Bacterial production without (top) and with (bottom) normalization to cell abundance over the incubation period for the five treatments (given in Fig. 1). Error bars represent the standard errors. The thin bars on the left of $T_0$ values indicate the 95% confidence interval at $T_0$ (colour codes as in Fig. 2 caption). Treatment abbreviations are provided in caption of Figure 1.**

The normalization of BP by the bacterial abundance brought up another perspective to these data (Fig. 5). Exposure to sunlight was no longer a significant factor, and differences between treatments appeared smaller, especially at

day 8 where the differences in BP were not significant (posthoc, P > 0.09). Nevertheless, it can be noted that the only treatment showing a continuously increasing trend was BL.

## 3.4 Carbon losses and gains during the experiment

Assessment of the carbon balance allowed to check whether the measured processes (DIC production, bacterial growth) were enough to describe the carbon transfers occurring in the incubation bottles (Fig. 6).

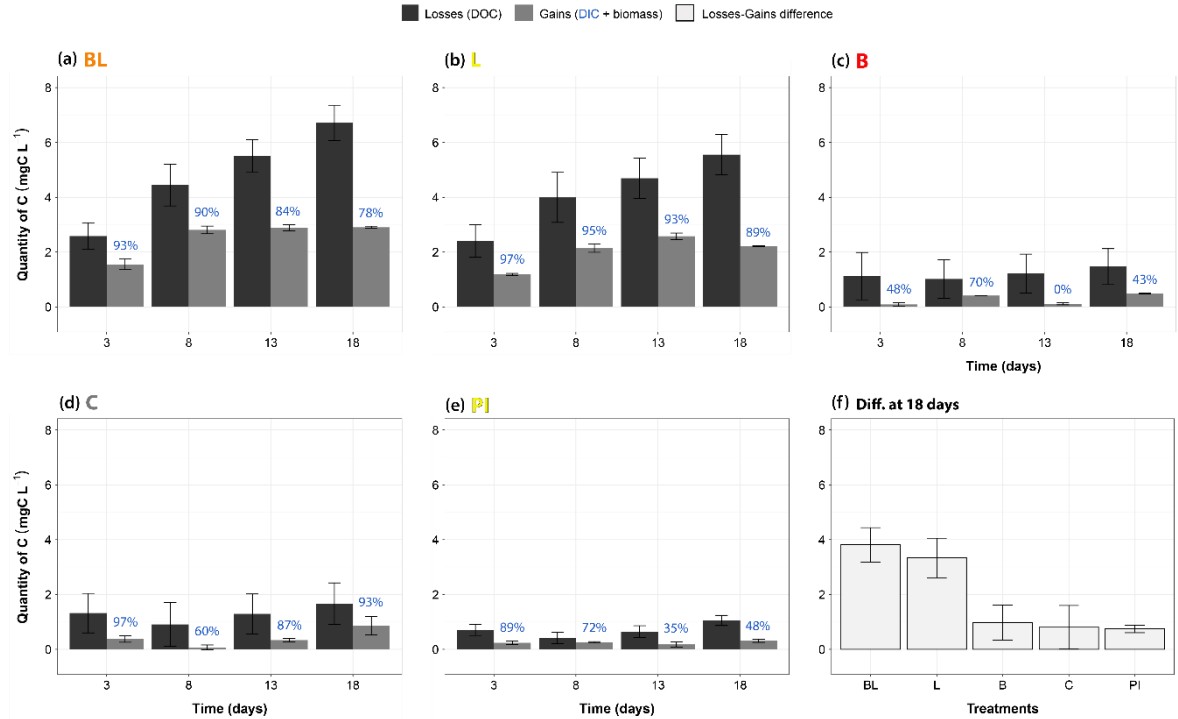

**Figure 6. Mass balance of measured carbon transfers during the incubation, categorized as "losses" and "gains". Panels (a) to (e) correspond to treatments BL, L, B, C and PI respectively, and present the mass changes of carbon relative to $T_0$ for days 3, 8, 13 and 18. Black bars represent the DOC losses (through sun mineralization, bacterial consumption, plus all other processes not directly measured), and grey bars represent gains of carbon as the sum of carbon increases in DIC and bacterial biomass. The proportion of gains as DIC is indicated in blue, above the bars. The bacterial biomass carbon gains were estimated for each time period using the highest BP value obtained for each treatment in order to evaluate the maximal potential carbon transfer into bacterial biomass. The panel (f) summarizes the difference between losses and gains for each treatment at the end of the incubation, potentially representing DOC flocculation but also other unmeasured processes like DOC conversion into carbon monoxide for the light treatments (see discussion). Error bars represent the standard errors. Treatment abbreviations are provided in caption of Figure 1.**

In all treatments and at all times, the measured carbon loss (as a decrease in DOC) was always higher (+58% to +1214%) than the measured carbon gain (as the sum of DIC and BP), even after maximizing BP in the calculation (i.e. by applying the highest rate obtained to all time periods of any specific treatment). BP gains thus represented on average 10% and 38% of the total gain measured, respectively for the light treatments (BL and L), and for the dark treatments (B, C and PI). Also, the uncertainty around the loss term was generally much higher than around the gain term (mean of 1.06 against 0.15 mgC L$^{-1}$). However, the difference between losses (in DOC) and gains (in DIC and bacterial biomass) was clearly larger for the two light treatments: on average 3.6 mgC L$^{-1}$ after 18 days of incubation, as compared to 0.8 mgC L$^{-1}$ for the three other treatments (Fig. 6f, Wilcoxon test, P < 0.0004).

Moreover, BL and L were characterized by a unique pattern in which carbon losses kept increasing, while carbon gains apparently reached a plateau from day 8.

## 4 Discussion

### 4.1 Sunlight as a driver of DOM mineralization

One of the main results of this study is the clear dominance of sunlight as a driver of DOM degradation and mineralization in the studied peatland thermokarst lake at the end of winter. In BL and L treatments, the same significant concurrent changes were observed: $CO_2$ was produced while DOC, CDOM and FDOM aromatic fractions declined, along with a general decrease in DOM molecular weight (increase of $S_{285}$). These changes were limited or absent in dark conditions.

Light absorption by CDOM cause a destruction of the chromophores. Optically, this translates into bleaching and the breakage of high-molecular weight molecules (Helms et al., 2008; Moran et al., 2000). When this photon-mediated oxidation is complete, $CO_2$ is produced and a corresponding carbon loss is going from DOC to gaseous state (Granéli et al., 1996). When the oxidation process is uncomplete, various moieties with different stages of oxidation accumulate in water, and their transformation does not result in DOC loss (e.g. Ward et al., 2014). The experiment here shows that both types of reactions occurred. About 4 mgC $L^{-1}$ were lost over 18 days through the direct action of sunlight (L treatment minus C, DOC loss rate of 0.22 mgC $L^{-1}$ day$^{-1}$). A certain fraction of this loss was certainly generated by the unexpected growth of bacteria in the L treatment, however a much larger fraction of this loss was likely due to direct photomineralization for two reasons. First, the DIC production was the highest over the first 3 days (0.38 mM day$^{-1}$) when bacterial regrowth was still minor. Second, when using the highest BP rate measured in the L treatment (0.54 µgC $L^{-1}$ h$^{-1}$, measured at day 8), we estimate that a maximum of 0.23 mgC $L^{-1}$ over the incubation period could have been lost through this biological pathway, representing about 6% of the DOC loss in L treatment. By comparison, 5 mgC $L^{-1}$ were lost in the BL treatment (after subtraction of C treatment, DOC loss rate of 0.28 mgC $L^{-1}$ day$^{-1}$). The difference between BL and L treatments can be attributed to the indirect action of sunlight favouring the bacterial consumption of DOC through the production of partially photooxidized biolabile substrates (at a conservative rate of 0.06 mgC $L^{-1}$ day$^{-1}$). Overall, we can consider that most DOC lost in BL treatment was triggered by sunlight, with a maximum of 78% resulting from its direct action (such as full photoconversion into $CO_2$) and a minimum of 22% resulting from its indirect action (such as bacterial consumption of photoproducts). The DOC loss in BL and L treatments measured over 18 days respectively represented 23% and 18% of the initial DOC concentration. We systematically subtracted the DOC loss of the C treatment, considering it would include all DOC losses unrelated to light or bacteria (e.g. through adsorption or dark flocculation: see below). Even if there was a slight regrowth of bacteria also in this control treatment, we considered it negligible as there was no measurable DIC production, and the maximum amount of carbon allocated to bacterial production over the incubation period was estimated at 0.05 mgC $L^{-1}$ (using a BP of 0.11 µgC $L^{-1}$ h$^{-1}$, measured at day 8), representing less than 1% of the DOC loss in BL treatment.

We expect DOM from northern peatland thermokarst lake to be very sensitive to sunlight degradation, at least at ice melt. Other studies on boreal waters are supporting this result. For example, the integrated $CO_2$ photoproduction was 3.5 times larger in spring than in summer for three boreal lakes (Vachon et al., 2016). However, Shirokova et

al. (2019), who incubated DOM from a Russian frozen peatland network reported the absence of measurable DOC loss over one month of sunlight exposure, and attributed this result to a dominance of refractory allochthonous DOM from soil and peat in this region. Laurion et al. (2021) similarly observed that DOM from organic-rich Arctic tundra ponds was poorly photomineralized, even though it was bleached. These differences suggest a variability in peatland DOM reactivity among regions, potentially related to different historical processing of the organic matter and thus, different molecular assemblages in DOM pools. Historical processing could differ prior to permafrost inception (Tank et al., 2020), or prior to sampling (Laurion et al., 2021). Also, the sunlight dose received (historical dose in situ, or during an experiment) needs to be closely taken into account when comparing studies. For instance, a major difference between Shirokova et al. (2019) and our experiment is related to the season when water was collected (late winter vs peaking summer), which likely has a strong impact on photoexposure history and thus on DOM photolability. Seasonal variations have to be further quantified to make more accurate predictions on the effects of winter contraction associated to climate warming.

### 4.2 The potential role of flocculation in the DOC loss

In our experiment, DOC losses under sunlight were not fully recovered in the form of DIC or bacterial biomass, and the relative gap between carbon losses and gains was increasing over time (Fig. 6). After 18 days, no more than 51% of the DOC losses were actually recovered in the form of DIC (from direct photomineralization and photo-stimulated bacterial respiration), which corresponds to about only 10% of the initial DOC concentration. This necessarily implies the existence of other light-mediated processes. Part of the unaccounted DOC loss could have been transformed into carbon monoxide (not measured in the present study), another common photoproduct that depends on the presence of certain DOM aromatic groups (Mopper et al., 2015; Stubbins et al., 2008), although this production would be low (a 1 to 15 ratio to DIC) according to Moran and Zepp (1997). Most importantly, the natural formation of flocs of organic matter (flocculation) could explain this gap. Several studies showed that sunlight is a triggering factor to the formation of particles and colloids, particularly in CDOM and iron-rich waters (Gao and Zepp, 1998; Helms et al., 2013; Mopper et al., 2015; Oleinikova et al., 2017; von Wachenfeldt et al., 2008). For example, von Wachenfeldt et al. (2008) reported that the synthesis of particulate organic matter from DOC of a sphagnum-dominated mire water almost reached 0.3 mgC $L^{-1}$ $day^{-1}$. It has also been shown that rising temperature can favour the transformation of DOC to flocs (Porcal et al., 2015; von Wachenfeldt et al., 2009). We indeed observed particles in filtered-water bottles but did not account for them quantitatively. In summary, photo-flocculation was likely occurring in our experimental water, which presented high DOC, CDOM and iron concentrations (Table S1) and was subject to high temperature peaks during the incubation (Fig. S1). Nevertheless, we cannot exclude that two technical issues leading to losses of the $CO_2$ produced could have also played a role in the gap between DOC loss and DIC gains: 1- dissolved $CO_2$ turned into gas phase and escaping when the incubation bottles were opened to collect DIC, although air bubbles were very small (1-2 mm of diameter), and 2- the storage delay of exetainer vials exceeding what was recommended by Faust and Liebig (2018).

The DOC loss under B treatment (bacteria and putative small grazers, 1.5 mgC $L^{-1}$) was similar to the one in the control treatment (5% of initial BA and no grazers, 1.7 mgC $L^{-1}$), suggesting that in the absence of sunlight, carbon losses were not associated to bacterial activity. In fact, the similar gaps between carbon losses and gains in B and C treatments (Fig. 6, not showing a specific temporal trend) could be attributed to the adsorption of DOM

molecules onto bottle walls (potentially quicker than flocculation). Helms et al. (2013) experimentally estimated that such adsorption was responsible for the loss of up to 0.08 mgC L$^{-1}$ day$^{-1}$ (on 550 mL quartz bottles). They also reported that this phenomenon was apparently strengthened by sunlight exposure, so this potential adsorption may have been even stronger in BL and L treatments. At last, bacteria-mediated flocculation could also have happened during dark incubations. This has been observed in water from oligotrophic boreal lakes in winter, where the bacterial activity would have favoured the formation of detrital flocs from CDOM (von Wachenfeldt et al., 2009). Our results suggest that flocculation may be important in peatland thermokarst lakes, and thus play a critical role in carbon burial. This has been highlighted as a main topic of interest related to the future of aquatic ecosystems affected by thawing permafrost (Vonk et al., 2015b). Flocculation should be systematically quantified during experimental assays testing on the role of sunlight in carbon cycling. Other studies corroborate the idea that boreal lakes could be strong carbon sinks through sedimentation processes, especially in ecosystems with high allochthonous humic DOM (Guillemette et al., 2017; von Wachenfeldt and Tranvik, 2008).

**4.3 DOM biodegradation and bacterial dynamics in the dark treatments**

The quantitative and qualitative changes of DOM in dark treatments were insignificant and comparable among B, C and PI treatments (Fig. 2), despite different bacterial community size and activity, notably between B and the two other treatments (Fig. 4 & 5). The data suggest that late winter DOM was refractory to biodegradation, at least over the incubation period and despite the oxygenation and temperature increase generated by the experimental setup. Communities were active in these treatments (BP > 0 and modest bacterial growth), but the insignificant DOC loss generated suggests a negligible consumption of carbon through bacterial growth and respiration ($CO_2$ production could not be detected). However, it is unclear what part of the DOM sustained bacteria activity as CDOM neither changed much (except for the small loss of CDOM observed in B treatment over the first 8 days, concurrent to an increase in BP; Fig. 2 and Fig. 5). This is different than results reported by Vonk et al. (2015a) for large streams and rivers, where biodegradation rates were relatively higher in winter than in summer, attributed to shifts in carbon sources and shorter hydrologic residence time in soils. However, there is a difference between lotic and lentic ecosystems in their connectivity with the landscape during winter, that is likely driving this difference in biodegradation. Moreover, it is possible that winter DOM would be biodegraded over longer time-scales. For example, Vähätalo and Wetzel (2008) reported that wetland-origin DOM, apparently biorefractory over 14 days, was actually reduced by half over 2.5 years of dark incubation. The importance of degradation over longer time scale becomes relevant in water bodies that have long residence time such as ponds.

Overall, we cannot exclude that the pre-experimental delay of four months could have further depleted the DOM pool of any labile material left at the end of a long winter, or modified the original bacterial community. Indeed, the temperature difference between water under the ice cover in March (0.5 to 2°C according to Matveev et al., 2019) and the storage cold chamber (4°C) could have been enough to stimulate the bacterial consumption of DOM. Bacterial activity is widely influenced by temperature, even by small differences (see Adams et al., 2010 and references therein). Nevertheless, the absence of a DOC decline between water collection and T$_0$ of the experiment (Table S1 and Table 1) supports the fact that DOC was recalcitrant to bacterial consumption, as observed during the experimental incubation in the dark. Actually, the DOC increased during storage, likely caused by natural or bacterial-triggered leaching from particles and colloids in the water, stored unfiltered. The largest change observed

during water storage was the production of FDOM, which has also been observed during the experimental incubation, and which we attributed to bacterial DOM transformation (see below). The reasons for CDOM and $SUVA_{254}$ declines are unclear but could both result from bacterial activity and preferential flocculation of the chromophoric fraction. More generally, it is possible that bacterial degradation in winter could be preferentially linked to the particulate fraction of organic matter, which was filtered out prior to the experiment. Free-living and particle-associated bacterial communities have indeed been shown to follow distinct seasonal patterns (Rösel et al., 2012). Therefore, it is possible that bacterial consumption of DOM in the 1.5 µm-filtered treatment (B) was underestimated both because of this 4-months delay and the removal of particles.

From an ecological perspective, the recalcitrance of late-winter DOM to biodegradation in this thermokarst lake is not surprising. It means that the system, at that period, would mainly concentrate molecules at the lower end of the reactivity continuum described by Mostovaya et al. (2017). This is supported by the initial properties of this DOM pool, strongly dominated by aromatic compounds with a typical terrestrial signature (high $SUVA_{254}$ and CDOM, only humic-like FDOM components with allochthonous features). As winter goes by, water gets more depleted in labile material, consumed first and foremost, while fresh inputs of carbon are impeded by the ice-cover, the frozen surrounding soils, and the absence of primary production in the lake and on land (Deshpande et al., 2016; Przytulska et al., 2016; Shirokova et al., 2021). Overall, this is consistent with studies on peatland and organic-rich wetland, generally reporting low bacterial growth efficiencies or low microbial DOM processing (i.e. Berggren et al., 2007; Laurion et al., 2021; Shirokova et al., 2019). By comparison, bacterial activity is clearly higher when supplied with fresh plant leachates (Shirokova et al., 2021). The abundance of low-quality substrates in peatland water may be related to the dominance of Sphagnum mosses at the study sites. These mosses are known to produce litter of poor organic matter quality that is decomposing slowly because their chemical structure holds large concentrations of phenolic, nonpolar and antimicrobial compounds (Turetsky, 2003).

Despite the absence of DOM consumption, microbial communities were relatively active and growing in the dark (Fig. 4 and Fig. 5). For example, the normalized BP in the B treatment was particularly high at day 8, and the BA substantially increased from $T_0$ in PI and C treatments. Dark treatments were also characterized by a notable increase in the signal of all fluorescent components, starting after 8 days of incubation (Fig. 3), just like what was observed during the pre-experimental delay (Table 1 and Table S1). This increase presumably happened also in the light treatments BL and L, noticed in the slowing declines of fluorophores (sometimes, even a small increase). Bacteria can produce FDOM, and this has been broadly observed in boreal and arctic lakes (e.g. Berggren et al., 2020; Laurion et al., 2021). The study of Berggren et al. (2020) on 101 boreal lakes suggests the reciprocal dependence between microbial processes and DOM optical properties. Considering the concurrent increase in BP and FDOM, we assume that bacteria were responsible for this synthesis of fluorescent molecules (similar assumption made in Logozzo et al., 2021, for example). The five humic components identified here had mostly been described as terrestrial, but our results suggest that they could also be produced by aquatic bacteria (although in a lake strongly influenced by eroding soils). It is interesting to note that there was no parallel increase of CDOM ($a_{320}$ decreased), suggesting that chromophoric non-fluorescent DOM would have been transformed into FDOM. Overall, bacteria have likely transformed DOM into different types of molecules (production of FDOM and potential other metabolic by-products) instead of strictly consuming it (that would have translated into a DOC decrease).

Lastly, it should be noted that the BP measured in the dark treatments is probably not representative of in situ late-winter production, where anoxia prevails and water temperature is close to 0°C. The experimental water was oxygenated during the filtration steps and the temperature regularly peaked around 30°C during the incubation (Fig. S1), exceeding the maximum temperature reached in situ in summer (18°C at the surface, Matveev et al., 2019). The bacterial production reported here was similar to what was measured from the same lake in summer

(Deshpande et al., 2016), while studies on other thermokarst lakes of the region found strong differences between summer and winter production (Roiha et al., 2015). Nevertheless, our results suggest that the bacterial metabolism was more limited by the quality of DOM than by the environmental conditions (see below), in accordance with results from Bižić-Ionescu et al. (2014) who showed that carbon rather than temperature was the limiting factor on bacterial growth in a temperate oligotrophic lake in winter.

**4.4 The photochemical stimulation of bacteria**

The exposure to sunlight had a positive impact on the bacterial dynamics of this thermokarst lake, such as noted for ice-wedge trough ponds in the High Arctic by Laurion et al. (2021). On one hand, when only a small proportion of the initial bacterial community was left (~5%, L treatment), sunlight exposure induced the exponential growth of these few bacteria, reaching about 10 times the abundance in the dark by the end of the incubation (C treatment;

Fig. 4). The BP also increased in the first week but dropped at the end of the experiment (no measurements between days 8 and 18; Fig. 5), possibly indicating that this population was reaching a stationary phase in response to a decline in the photoproduction of high-quality substrates, expressed as a slowing rise in BA. On the other hand, sunlight induced a more persistent rise in BP for BL treatment, although the difference in BP was apparently linked to differences in population size (see normalized BP, Fig. 5b). Yet, the increase in BP did not translate into rising

biomass in this treatment. This could be explained by the grazing pressure exerted on bacteria by small protists that were able to pass through the nominal 1.5 µm filtration step. Nutrient enrichment experiments realized on summer water from the same lake also showed that BP was maintained or increased while BA dropped (Deshpande et al., 2016) suggesting an important top-down control of bacterial populations in this lake (Bégin and Vincent, 2017; Przytulska et al., 2016). Complex mechanisms involving the viral population could also control the size of

the bacterial community facing important changes such as in spring. Girard et al. (2020) demonstrated that communities in SAS2A lake were active and seasonally shifting in response to the population dynamics of host organisms. Interestingly, this potential viral control would not have acted on the rising population observed in L treatment, potentially linked to the inhibiting action of UV radiation on viral activity.

The observed intensification of protein and cell synthesis under sunlight exposure indicates a photochemical

conversion of part of the DOM into substrates readily available for bacterial growth, as it was previously described in the literature (e.g. Moran and Zepp, 1997). Most common photoproducts are organic acids, but other low molecular weight compounds containing essential nutrients could also be involved (Pullin et al., 2004; Vähätalo et al., 2003; Wetzel et al., 1995). For example, Xie et al. (2012) reported the photoproduction of ammonium from DOM. The higher BP in BL treatment clearly indicates that the bacterial community was benefiting from the

photoproduction of essential compounds. However, the insignificant difference in DIC production between BL and L treatments, especially between 0 and 3 days when the BA in L was still negligible, suggests that most $CO_2$ was directly photoproduced and not resulting from respiration of this fast-growing population.

Results also suggest that the initial sunlight exposure of 2 days in PI treatment was not sufficient to substantially stimulate microbial metabolism. The fact that the initial community was small (inoculum) is not likely to be the reason, because L treatment demonstrated that the initial size of the community was not a strong determinant of its activity in the following days. We hypothesize that the limiting factor was rather the limited amount of photolabile compounds produced over the 2 days of sunlight pre-exposure. CDOM decreased by 14% and $S_{285}$ increased by 10%, indicating that photodegradation happened (Table 1), but photoproducts may have been consumed rapidly and the stimulation turned down as soon as the water was placed back in the dark. The superposition of C and L curves in the first 3 days could also indicate that a minimum length of sunlight exposure was needed to produce a boosting effect on bacteria. This lag could reflect the adaptation time needed by the community to respond to a change in the quality of the carbon resource. Ward et al. (2017) demonstrated that shifts in substrates availability were driving deep shifts in bacterial gene expression and composition, translating into time lags in bacterial metabolism. These effects would be occurring in a water column intermittently mixing, generating a complex DOM pool containing molecules photodegraded at various degrees (Matveev et al., 2019). Achieving sterility is a difficult task and we argue that BA should always be controlled in such experiments (Logozzo et al., 2021 and references therein). Not considering this could lead to interpretation biases. Filtration artefacts can impact short-term biolability assays through their effects on bacterial abundance, composition and predation as demonstrated for waters draining from peatlands in the Netherlands (Dean et al., 2018). Although we achieved to reduce the initial BA by 95% in L treatment, it took about 8 days for these bacteria to overtake the abundance observed in BL treatment. Although the effect of sunlight could not be isolated completely, the experiment clearly demonstrated the efficient role of sunlight on bacterial growth and carbon mineralization. We further noted that cells with a higher DNA content seemed to be particularly abundant in the light treatments at the end of the incubation (Fig. S3 showing higher proportions of MNA and HNA populations at 18 days). A taxonomic characterization of these bacterial populations appearing on the cytograms (using a cell sorter for example, see Fig. S4) is needed to better understand the effects of sunlight on the bacterial cycling of permafrost carbon and composition structure of the communities.

**4.5 Significance at the lake scale**

The strong impact of sunlight reported here needs to be considered in the specific context of thermokarst lakes. In such humic lakes, only surface layers receive photochemically active photons. For example, Vähätalo et al. (2000) calculated that 90% of photomineralization happened in the top meter of a humic lake, with UV radiation contributing to 77% of this mineralization. Similarly, the study from Koehler et al. (2014) based on 1086 Swedish lakes concluded that 95% of the depth-integrated DIC photoproduction happened in the upper 0.8 m of the water column. Attenuation coefficients were estimated from downwelling irradiance profiles realized in SAS2A in August 2016 (unpublished data) with a multispectral radiometer (Satlantic Inc., now owned by Sea-bird scientific). In July, 90% of incident photons were attenuated at 5, 14 and 23 cm respectively for wavelengths 380, 443 and 491 nm. This is considerably shallower than the photoactive depth mentioned above for Swedish lakes, but it is important to note that our study lake is extremely humic by comparison to the majority of the lakes in the database used by Koehler et al. (2014): only 25% have an absorption coefficient at 305 nm above 89 $m^{-1}$, against a value of 143 $m^{-1}$ for SAS2A lake at $T_0$. Therefore, most of direct photomineralization in SAS2A is occurring in the first

centimetres of the water column. In our experiment, degradation was constrained to a 7.3 cm-depth (diameter of bottles floating under a thin layer of transparent tap water) which makes it representative of a typical exposure at the very surface of the lake. Previous studies indicate that thermokarst lakes in this area are strongly stratified most of the year, with brief mixing periods in spring and autumn which depend on lake morphology (Matveev et al., 2019). Notably, the small size of these lakes and their high DOM content impede on a complete spring turnover, where only top layers get mixed for a few days until summer stratification establishes. Partial mixing then occurs on a daily basis at night. This mixing regime generates conditions where sunlight can only reach surface layers, but partial mixing at night likely brings fresh DOM at the surface on a diurnal basis. Thus, the experimental exposure of the same DOM molecules for 18 days may have underestimated the photodegradation potentially taking place in situ, where substrate limitation is less likely to occur. On the other hand, incident irradiance during the experiment down south was 30 to 103% higher than at the study site (depending on the year – Fig. S1), likely causing an acceleration in photochemical decomposition over the first days of the incubation, but this was followed by the plateauing of certain proxies (e.g., DIC, bacterial growth in L treatment) suggesting the limitation of substrates in the absence of mixing renewal.

At last, the study by Deshpande et al. (2016) highlighted the importance of attached bacteria in SAS2A lake, where more than 70% of BA and BP was associated to particles larger than 3 µm. As most particles were removed in the present experiment (filtration either through 1.5 or 0.2 µm), it is likely that bacterioplankton respiration, growth and DOM degradation were underestimated. Moreover, our experiment provides information on DOM biodegradation potentially happening in the upper oxygenated layers, but it does not address bacterial activity in deeper anoxic layers of the lake. To fully account on the importance of dark bacterial mineralization relative to photomineralization, experiments will have to include particle-attached flora and respect the anoxic conditions prevailing in these systems. The present study can only underline that photomineralization is an efficient process accelerating DOM processing in epilimnetic waters of thermokarst lakes.

**5 Conclusion**

Our work suggests that winter DOM collected from a thermokarst lake in a subarctic peatland is having a high potential of photoreactivity at ice-off, and that the bacterial community originating from the water was strongly stimulated by sunlight. About 5.0 mgC $L^{-1}$ (23% of initial DOC concentration) were lost over 18 days when bacteria and sunlight were both at play. Results indicate that (1) direct effects of sunlight (78% of this DOC loss) were much larger than indirect effects (22% of the DOC loss, stimulation of bacterial consumption through the synthesis of labile photoproducts), (2) photomineralization ($CO_2$ production by direct or indirect effects) did not account for more than 51% of the DOC losses (~10% of initial DOC concentration), and (3) most of the remaining DOC loss was likely associated to photo-flocculation. The positive effect of sunlight on the bacterial community was particularly demonstrated by the outstanding regrowth of the population left in L treatment (even after the 0.2 µm filtration). Without sunlight stimulation, no detectable DOC or CDOM losses could be attributed to the sole action of bacteria, yet they remained relatively active in the dark and produced FDOM. The highly-coloured aromatic DOM found in these lakes at the end of winter was quite refractory to biodegradation but particularly sensitive to sunlight. Our work thus suggests that sunlight is an important mediator of $CO_2$ emission and carbon burial in peatland thermokarst lakes after the ice cover melts. The next step will be to quantify this cycling at the

lake scale by taking into account the natural conditions prevailing in spring, including local irradiance, temperature
and oxygen concentration. A direct quantification of DOM flocculation and its mediation by sunlight are also
needed. Finally, the inclusion of particle-attached microorganisms and the characterization of bacterial
communities during this seasonal transition (from dark anoxic conditions under the ice cover to varying exposure
to sunlight and oxygen in the diurnally-mixed surface layer of the lake) would help to improve our understanding
of climate change effects on the carbon cycling in this important class of lakes.

**Data availability**. The dataset of the experience and related temperature-irradiance metadata are deposited in
NordicanaD at the DOI: 10.5885/45759CE-9ED4BB6AE585446C (Mazoyer et al., 2022).

**Supplement link.** This article contains a supplement information.

**Authors contribution.** The field campaign was conducted by IL and MR. FM designed and performed the
experiment with the help of IL. FM analyzed the data and wrote the manuscript under the supervision of IL and
MR.

**Competing interests.** The authors declare that they have no conflict of interest.

**Acknowledgements.** We thank Martin Pilote, Alex Matveev, Joao Canario, Alice Lévesque, and the Cri guide
Thomas Shem for their help in the field, Gilles Guérin for his logistical assistance to set up the experiment on the
lab roof, Jérôme Comte for the use of his flow cytometer and his helpful advices, along with Audrey-Anne Boutin
for her help with cytometric analyses. We also thank Maxime Wauthy and François Guillemette for their help on
building the PARAFAC model, and Mathieu Cusson for his advices on the statistical analyses. We are grateful to
the ongoing support by the CEN for field work and data accessibility. Irradiance data for Québec City were kindly
provided by the electrical engineering and computer engineering department of Laval University. Irradiance data
for the SAS site belong to Florent Dominé (Takuvik Joint International Laboratory, Laval University). At last, we
thank our two anonymous reviewers, Brent Dalzell, along with Liudmila Shirokova, for their constructive
comments. The project was supported by a seeding grant from the CEN Hudsonie21 program, NSERC discovery
and northern supplement grants to IL and MR, along with a scholarship from EnviroNorth CREATE program to
FM.

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
