# Peer review of "The dominant role of sunlight in degrading winter dissolved organic matter from a thermokarst lake in a subarctic peatland"

_Biogeosciences, 2022_

## Author Response (AR1)

**Answer to comments from Reviewer 1**

We thank the reviewer for his (or her) time evaluating the paper and for his (or her) precise and well-argued comments. Please, see our answers below.

**Shortcoming 1**: *The authors aimed to test the reactivity of DOM after ice-off in a lake, but had an altered DOM composition and possibly different bacterial community composition during their experiments compared to the water originally present in the lake. It is likely that the changes in DOM composition (shown by the substantially lower SUVA254 value and higher Fmax value) measured between the initial water sampled (Table S1) and water at the start of the experiments (Table 1) were due to bacterial mineralization of DOM during the 4-month storage period of the unfiltered lake water. The authors found that fluorescent DOM increased with bacterial production during their experiment. Then, previous work from these authors showed that bacteria in similar lakes degrade aromatic DOM (Laurion et al, 2021).*
*Many papers have shown how bacterial community composition and activities change to adapt to new DOM compositions (Crump et al, 2003, Dinasquet et al, 2013, Logue et al, 2016). If bacterial community composition did change to adapt to the altered DOM composition, then the bacterial production and mineralization measured during the experiment does not reflect what is taking place in the lake water. I think that the authors should include these limitations in their discussion because they impact the authors' conclusions on how much sunlight and bacteria can mineralize DOM to CO2 following iceoff in thermokarst lakes.*

**REPLY**: We first want to underline that lakes at SAS site are very different from the ones at Bylot site (Laurion et al., 2021). Moreover, this study on Bylot was done on DOM collected after a dry period in July, but see reply to shortcoming 2.
We agree with reviewer 1 that this delay, due to logistical reasons, is not ideal, and that this water is not completely representative of the DOM and bacterial community at ice-off. We already underlined in the manuscript that winter-like conditions (~4°C, darkness, no oxygen) were artificially prolonged by 2 months in the refrigerator as compared to the ice-break occurring mid-May (Matveev et al., 2019), but we clarify this point further (lines 65-66, 105-118, 519-534). For example, we know that under natural conditions, phototrophic production can start again as soon as light is becoming more available under the ice, changing the DOM pool and the bacterial community before spring melt (Bertilsson et al., 2013). The purpose of the study was to explore the degradation potential of winter DOM pool that is rich in recalcitrant carbon after a long dark period without external inputs, rather than the degradation potential of spring DOM pool that can already be enriched in autotrophic DOM and other inputs from the ice-melt period.
The decrease in SUVA and increase in FDOM seem to have indeed continued during this prolonged period, but we think this should not be considered as an experimental flaw. Moreover, we think that changes in bacterial assemblages were already adapted to a DOM pool with limited lability after more than 5 months under an ice cover (from the ice formation mid-October to sampling end of March). We agree that the community has likely changed from in situ conditions in March until T0 of the experiment, but more because of its transfer into a container than because of a prolonged winter. We assumed that most changes would have occurred relatively quickly after water collection, and that prolongating by 2 months would not change much on the story. Any experimental set up has effects on microbial assemblages, and only a few people doing experiments can overcome such effects if the logistics is simpler. We see no reason why the DOM-bacteria tandem would evolve that differently in the container than it would evolve under the ice in the deep of the winter. The main structuring physico-chemical conditions were very close: 4°C, darkness, no oxygen. These are likely the most important characteristics defining the habitat and influencing the metabolism (Bertilsson et al., 2013; Jansen et al., 2021).
The literature proposed by reviewer 1 mainly points out how organic matter source and quality drive shifts in bacterial communities. For example, Crump et al. (2003) noted that there are persistent

50 populations throughout the year and that the transient populations are linked to spring terrestrial inputs and summer phytoplankton development. Actually, the results of this work even suggest that, in the absence of a strong perturbation in the DOM pool (strong inputs suddenly coming or suddenly stopping), changes in the bacterial community should be limited. Between water collection and the start of the experiment, the collected water did not receive any DOM inputs, so we think that changes
55 in the bacterial community were limited, as discussed in the proposed articles.

**Shortcoming 2:** *The authors did not compare their findings to Laurion et al. (2021), which studied DOM mineralization in other lakes in the region with a similar experimental design. Because Laurion et al (2021) conducted the sunlight exposure and bacterial incubation experiments sooner after water sample collection, the authors could benefit from discussing how their results compare and how they*
60 *might be due to differences in the experimental design alone (not due to different study sites).*

**REPLY**: We do not think that these two studies are easily comparable. These ponds have in common to be wetland thermokarst systems, but there are many other differences. Bylot Arctic ponds (73°N) from Laurion et al. (2021) are diverse, with clear to brown-colour and DOC content generally around 10 to 13 mgC $L^{-1}$. Local soil is perennially frozen (continuous permafrost), which limits hydrological
65 exchanges. On the other hand, SAS ponds (like SAS2A in our study) are subarctic ponds (55°N) lying in ombrotrophic bogs, with ice remaining only in the heart of palsas (sporadic permafrost). Waters are black-coloured and much richer in DOC (~22 mgC $L^{-1}$ here). Surrounding unfrozen soils are flooded, which facilitates exchanges. Soils from Bylot and SAS do not have the same age nor the same history, and the surrounding vegetation is different. At last, the experiment from Laurion et al. (2021) was
70 made on summer water, containing DOM inputs from phytoplankton, macrophytes and the surrounding land (although after a dry period with limited inputs). On the contrary, our experiment was done on winter water, deprived from summer inputs, which makes a very big difference. The novelty of this study has been further underlined (lines 64 and 67-68).

*Lines 18-21: It is incorrect to report that 18% of DOC was directly lost over 18 days of sunlight exposure*
75 *when the higher abundances of bacteria later in the experiment could have contributed to that mineralization. The authors do a good job discussing how much DOM mineralization could have come from sunlight and bacteria, so this should be clarified in the abstract too.*

**REPLY**: Thanks to the reviewer for bringing this up. We have adjusted the text accordingly stating that "Up to to 18%" was lost and we have made it clearer that there had been a bacterial regrowth (lines
80 18 and 20-22).

*Lines 106-108: The sentence "Indeed, DOM variables … and aromaticity)" needs to be revised to accurately report the results. Tables S1 and 1 show substantial changes in DOC, SUVA254, and the fluorescent DOM components during the 4-month storage period. Statistical tests should be used to report the statistical significance of those differences.*

85 **REPLY**: Unfortunately, these differences cannot be statistically tested because we do not have the adequate replication for that. The data give an idea of the qualitative change that happened, but we agree that a quantitative appreciation of these changes was missing. We now report more accurately these changes and added more justification as mentioned above (lines 108-118).

*Table 1: Define all of the abbreviations in the caption.*

90 **REPLY**: We redefined explicitly the treatments in the caption as suggested.

*Lines 144-145: Can you cite a paper where this is reported?*

**REPLY**: We added a reference were the same type of bottles have been used to carry out photodegradation experiments and showing Teflon bottle transmittance spectra (lines 157-159).

*Line 483: It is not accurate to report that the DOM in the lake prior to ice-off is refractory to biodegradation when there were 4 months before the experiment when bacteria could have been mineralizing the most labile DOM to CO2."*

**REPLY**: When the water was sampled at the end of March, it was already after more than 5 months of winter conditions, but it is true that we cannot exclude the fact that there could have been labile molecules left by then. However, we do consider that this sentence is accurate considering that it derives from the biodegradation experiment results (and we say "the data suggest"). Nevertheless, we now address the point raised by the reviewer lower in the discussion (next paragraph; lines 519-534).

*In general, I think the manuscript would read better if each treatment were spelled out rather than abbreviated.*

**REPLY**: If possible, we would like to maintain these intuitive abbreviations (L for light, B for bacteria), since they help to make the text lighter.

**Cited References**

Bertilsson, S., Burgin, A., Carey, C. C., Fey, S. B., Grossart, H. P., Grubisic, L. M., Jones, I. D., Kirillin, G., Lennon, J. T., Shade, A. and Smyth, R. L.: The under-ice microbiome of seasonally frozen lakes, Limnol. Oceanogr., 58(6), 1998–2012, doi:10.4319/lo.2013.58.6.1998, 2013.

Crump, B. C., Kling, G. W., Bahr, M. and Hobbie, J. E.: Bacterioplankton community shifts in an Arctic lake correlate with seasonal changes in organic matter source, Appl. Environ. Microbiol., 69(4), 2253–2268, doi:10.1128/AEM.69.4.2253-2268.2003, 2003.

Jansen, J., MacIntyre, S., Barrett, D. C., Chin, Y. P., Cortés, A., Forrest, A. L., Hrycik, A. R., Martin, R., McMeans, B. C., Rautio, M. and Schwefel, R.: Winter Limnology: How do Hydrodynamics and Biogeochemistry Shape Ecosystems Under Ice?, J. Geophys. Res. Biogeosciences, 126(6), 1–29, doi:10.1029/2020JG006237, 2021.

Laurion, I., Massicotte, P., Mazoyer, F., Negandhi, K. and Mladenov, N.: Weak mineralization despite strong processing of dissolved organic matter in Eastern Arctic tundra ponds, Limnol. Oceanogr., 66(S1), S47–S63, doi:10.1002/lno.11634, 2021.

Matveev, A., Laurion, I. and Vincent, W. F.: Winter Accumulation of Methane and its Variable Timing of Release from Thermokarst Lakes in Subarctic Peatlands, J. Geophys. Res. Biogeosciences, 124, 3521–3535, doi:10.1029/2019JG005078, 2019.

**Answer to comments from Reviewer 2**

We thank the reviewer for his (or her) time evaluating the paper and for his (or her) detailed report. Please, see our answers to your well-argued comments below.

***Time between sample collection and initiation of the incubation***. *Over four months elapsed between initial sample collection and the beginning of the incubation study. While the authors point out that the sample was stored in the dark at 4 °C, this still represents a dramatic shift from the sampling environment (0.5 °C just below the ice). Even under refrigerated conditions, four months is a long time for aquatic microbes to make use of the most labile portions of the aquatic DOM. It is likely that comparatively low microbial activity observed during the incubation may be at least partially due to the fact that this sample essentially underwent a four month microbial incubation prior to the beginning of the official experiment. This needs to be explored and results need to considered within this context. On a related note: it appears from Matveev et al., 2019 that the sample from SAS2A was collected on 19 March while the present paper states it was collected on 24 March. Please verify that the reported sampling date is correct.*

**REPLY**: The goal of this study was to investigate the fate of winter DOC pool during spring time, when oxygenation, sunlight and higher temperatures are coming back. As replied to reviewer 1, we agree that this delay, due to logistical reasons, was not ideal. However, we consider that the prolongation of these winter-like conditions is of 2 months rather than 4, because the ice-break naturally happens mid-May (Matveev et al., 2019). Yet, it is true that bacteria continued to transform DOM during 2 supplementary months, as already underlined in the first version of the manuscript (lines 102-106). Therefore, the changes in DOM properties during the delay (now described and justified more explicitly; lines 108-118) were potentially stronger than what naturally occurs in situ until spring melt. We also agree that the difference in temperature under the ice cover (between 0.5 and 2°C) and the environmental chamber where the container was kept (4°C) is not negligible and could have accelerated the bacterial transformation of DOM and potentially changed the community composition. This is now pointed out in the revised manuscript (lines 519-524). However, Adams et al. (2010) for example found that only 4 OTUs out of 79 were unique to the temperature of 2°C in the bacterial community from an Arctic lake. We assumed that all bacterial species were still present in the container water, but potentially in different proportions as in the lake at ice melt. This is always the case for container water kept to make experiments, probably already after a few days; this is not unique to water kept for months.
We agree that the experimental biodegradation could be underestimated because of this longer delay, and we now have addressed this more extensively in the manuscript (lines 519-534). However, we think that the absence of DOC consumption during this delay (in fact there was an increase in DOC by 15%) supports the claim of recalcitrance. Other DOM changes happened (in $SUVA_{254}$, CDOM and FDOM), but we expect these changes to be similar to changes occurring under the ice in dark and cold conditions, but simply extended in length. We cannot tell if they resulted from the action of the bacterial community, from flocculation, or from a combination of both.
About the date issue: samples used for the Matveev et al. (2019) study were indeed collected from the 19[th] March, but the water brought back in the carboy for our specific study was collected separately, at the end of the sampling campaign, on the 24[th].

***Incoming Solar Energy***: *If the Teflon bottles are light diffusers, then solar radiation measured outside the bottles is not representative of the energy experienced by the samples. You may be able to apply a simple correction by placing a pendant or pyranometer inside a Teflon bottle (or cut a bottle to make a Teflon cover) – this will be a more useful measure of the solar radiation that reached the sample.*

**REPLY**: As replied to reviewer 1 above, we added some justification about the use of Teflon, which is transparent to wavelengths of interest (lines 157-159). We hope it will satisfy this request.

***Challenges with sample filtration***. *(lines 120-131) It is unclear why different filtration schemes were used when preparing water for the experiment (e.g., 0.2μm Tuffryn vs a two-step process with a 0.7 μm glass fiber filter and 0.2 μm cellulose acetate filter.) Please explain the sample preparation more clearly, perhaps with a flow chart.*

175

**REPLY**: We understand your comment and tried to clarify this further (lines 140-144). We had to use both techniques to obtain a 0.2-μm filtrate for pure logistical reasons (shortage of filtering capsules after the ones planned started to clog). We describe honestly the steps, but do not want to distract the reader with these details into a flow chart.

180 ***Closing the DOC mass balance****. The authors observed a large mismatch (58 to 1214%) between DOC lost and the concomitant gain in DIC and bacterial production. They attribute this to DOM flocculation which was noted, but unfortunately not quantified. In the absence of supporting measurements, the authors probably don't need to spend so much time discussing flocculation and simply admit they don't know.*

185 *Additional considerations could include: DIC outgassing to $CO_2$ during the incubation. It is possible that some DIC was out of solution in the form of an air bubble in the bottle. This DIC would not have been accounted for. Was this observed or checked? $CO_2$ loss during the two-month period of sample storage between sampling and analysis. While sample storage in exetainer vials has generally been reported as stable, it has been demonstrated that $CO_2$ concentrations were up to 14% lower than expected in*

190 *vials that had been stored for 84 days.*

**REPLY**: We acknowledge that we have no direct measurements of flocculation, but we would not be the first to rely on indirect observations to discuss results! We really think that this paragraph deserves to remain in the discussion. First, because we have been systematically observing flocs in filtered water originating from SAS peatland lakes, under different circumstances of sample preservation. Second,

195 we think that it is relevant to raise awareness of the readers on measuring flocculation when investigating degradation of DOM in waters rich in aromatic components. Flocculation is an overlooked process that may be challenging to measure on small volumes, but we think it really deserves attention. Nonetheless, we toned down the subtitle (line 464) and added to the revised version of the manuscript these two other potential factors that could have contributed to the DOC-DIC gap (lines 481-485).

200 Notably, the air bubble in the incubation bottle at sampling was quite small (1-2 mm in diameter), so we suspect this was a negligible loss. We thank the reviewer for underlining this paper on $CO_2$ loss during storage in Exetainers, which will be relevant to researchers using them.

***Line 8 (and elsewhere)***: *suggest rewording "retroaction loop" with "positive feedback loop".*

**REPLY**: This was changed accordingly and throughout the manuscript.

205 ***Line 18***: *"full mineralization to $CO_2$" implies that the entire DOC pool has mineralized; this is not consistent with your data.*

**REPLY**: With the word -full- we meant the complete chemical conversion from the original DOM composition to $CO_2$, which applied to a portion of DOC. Full was not intended to mean that the full amount of DOM would have been mineralized to $CO_2$ but instead that a fraction of the DOM pool was

210 completely mineralized to $CO_2$, because $CO_2$ increased in the light treatment (and from the beginning of the incubation when bacterial regrowth was minimal). A second fraction of the DOM pool underwent photodegradation that did not lead to mineralization, a third fraction putatively flocculated, and one last fraction was probably not sensitive to photodegradation at all. We removed the term "full" to avoid any confusion (also raised by Dr L. Shirokova) and the sentence now reads:

215 "We demonstrate that sunlight was clearly driving the transformation of the DOM pool, part of which went through a complete mineralization into $CO_2$."

***Line 22***: *replace "undirect" with "indirect".*

**REPLY**: Changed accordingly.

*Line 23: "outstanding boosting factor" is awkward wording, please find alternative wording.*

220     **REPLY**: We replaced this expression by "considerably stimulated" (now line 20).

*Line 84: you refer to Fig. 7 of Vincent et al. (2017); perhaps you can include this figure in the supplemental information.*

**REPLY:** We would prefer not adding it and just keep it mentioned as it is, because it is not directly related to the experiment in itself. It is a reference for readers who would like to obtain more
225     information on the study site.

*Line 86: Field sampling – just refer to the date the sample was actually collected (was it 19 March or 24 March?).*

**REPLY**: Actually, both dates are correct. The field data collection was carried out on the 19[th] and 24[th] (see table S1), while the water for the experiment was collected on the 24[th] (it is specified at the end
230     of the paragraph). That is why we introduced the paragraph with that full period.

*Lines 144-145: provide a reference for the light-filtering properties of Teflon.*

**REPLY**: We added this information, as mentioned above (lines 157-159).

*Lines 211-213: Why include the unpublished data in the PARAFAC analysis?*

**REPLY**: An extra dataset involving another experiment at the same site was added to our dataset in
235     order to develop and validate a PARAFAC model more easily, but this other experiment is not yet published. According to Stedmon and Bro (2008), having a minimum of 100 samples indeed generally makes the component extraction more efficient during the validation steps. We clarified that in lines 226-228.

*Line 248: replace "unfrozen" with "thawed".*

240     **REPLY**: The comment refers to a sentence talking about water samples that were melted at room temperature. We replaced unfrozen by melted in the revised manuscript. It is probably better to keep thawed for solid material.

*Lines 294-303: Please provide more supporting references in your discussion of the fluorescence results.*

**REPLY**: We added some examples of supporting studies in this paragraph, but all supporting references
245     are provided in Table S2 (now explicitly specified in the manuscript, lines 310-317).

*Figure 6: Please show the DIC and Biomass as separate portions of the bar chart (stacked to show the total).*

**REPLY**: Since the percent into biomass is quite small it would not be seen easily as stacked bars (especially for treatments in the dark), so we instead added the percent represented by DIC over each
250     bar.

*Line 427: "… carbon canalized to bacterial production…" I think you mean to say "… carbon allocated to bacterial production…".*

**REPLY**: Changed accordingly.

*Lines 540-541: These details about filter preparation and problems with cracking need to be presented*
255     *in the methods section.*

**REPLY**: We transferred this information to the methods section, as suggested (lines 138-140).

*Lines 577-580: This material about the DNA content of cells should be excluded from this paper.*

**REPLY**: We do not understand the reason why the reviewer asks for this exclusion since the DNA content of cells provides indications on the structural composition of bacterial communities. This is now clarified in the manuscript (lines 623-628). We also changed Fig. S3 to better illustrate this structural change in the bacterial composition (providing the three populations in percent composition). With this new figure, we can clearly see that 0.2 µm-filtration eliminated MNA and HNA bacteria (in C and L), but when kept in the dark (C), the structural composition of the community converged to the same as in 1.5 µm-filtered samples, while the community exposed to sunlight (L) became much richer in MNA and HNA. We would like to keep this interesting result.

**Cited references**

Adams, H. E., Crump, B. C. and Kling, G. W.: Temperature controls on aquatic bacterial production and community dynamics in arctic lakes and streams, Environ. Microbiol., 12(5), 1319–1333, doi:10.1111/j.1462-2920.2010.02176.x, 2010.

Matveev, A., Laurion, I. and Vincent, W. F.: Winter Accumulation of Methane and its Variable Timing of Release from Thermokarst Lakes in Subarctic Peatlands, J. Geophys. Res. Biogeosciences, 124, 3521–3535, doi:10.1029/2019JG005078, 2019.

Stedmon, C. A. and Bro, R.: Characterizing dissolved organic matter fluorescence with parallel factor analysis: a tutorial, Limnol. Oceanogr. Methods, 6, 572–579, 2008.

**Answer to comments from Dr Liudmila Shirokova**

We are grateful to Dr Liudmila Shirokova that reviewed our manuscript, additionally to the two reviewers appointed by the editor. We thank her for her positive judgement and answered all her relevant comments below.

*In the Abstract, there is some self-contradictory: 18 % efficiency (L 18) is not full mineralization of the DOM pool.*

**REPLY**: We erased the term "full" in the sentence, which was probably leading to confusion (also brought by reviewer 2). In the first sentence we want to express that 1) sunlight drives the transformation of the DOM pool, and 2) part of this DOM pool ends up mineralized to $CO_2$. In the following sentence, we are entering into treatment descriptions. We simply start by expressing that 18% of the initial DOC disappeared in the light treatments. We hope this is now sufficiently clarified.

*L18 & L 24-25 again, half loss and full mineralization are not the same things.*

**REPLY**: We hope that the deletion of the term "full" clarified the apparent contradiction.

*L42 Here, a reference is needed. Note that sediment respiration, soil input and groundwater discharge are also important drivers."*

**REPLY**: We added two references to support this (line 44). We agree that inputs from soils and groundwater discharge may also be an important source of $CO_2$ to pond water, but we prefer not to distract the reader with these aspects here.

*Methodology: The change of redox conditions between sampling and storage – partial oxidation after aeration during sampling should be discussed.*

**REPLY**: Unfortunately, we did not follow dissolved oxygen concentration in the water after collection, neither during storage time. Dissolved oxygen may have slightly increased at sampling, although water was gently collected with a thin-layer sampler connected to a peristaltic pump (Matveev et al., 2019). However, results from Folhas et al. (2020) suggest that the study lake (SAS2A) is particularly difficult to oxygenate: ice drilling in other lakes of the site caused oxygen concentration to increase in the surface layer to 20-25 mg $L^{-1}$, but it did not go above 3 mg $L^{-1}$ in SAS2A. Also, since the container was full and kept closed in the cold chamber until the experiment started, we think storage conditions were similar to lake conditions in terms of anoxia. We added these arguments in the revised manuscript (lines 111-118).

*Table 1 Please add the pH value and specific conductivity.*

**REPLY**: Unfortunately, these variables were not measured during the experiment, only in situ.

*Detailed description of experimental setup is highly appreciated!*

**REPLY**: We thank you for this positive comment. We think these details are often missing.

*L132-133 The reason for this delay is not totally clear. Why this experiment was not run from 30 May to 18 June, given that the light conditions and temperature in summer are not the same as in early spring."*

**REPLY**: The reasons for the delay are totally logistical; time to plan and set up everything, along with coordination for other fieldwork. We added this precision in the revised manuscript (line 104). The ideal situation would have been to carry out the incubation directly in the field at ice melt to account for local irradiance and temperature, but we could not plan this way as the field station is remote, expensive to access and not equipped as down south. Although not ideal, this delayed experiment is

what we have to offer, and we still believe it provides interesting insights on the susceptibility of winter DOM to photodegradation. We tried to take this into account afterwards by providing and comparing irradiance and temperature in the field. Yet, an advantage of carrying out the experiment close to the laboratory was to perform DOM analyses very quickly, allowing to capture fast-cycling DOM (which may be lost even in filtered samples kept for a few days/weeks).

*L145 A reference for UV-transparency of FEP bottles is needed.*

**REPLY**: Please see what we added in response to this point also raised by reviewer 1 and 2 (lines 157-159).

*L211-212 Unclear, what kind of data are discussed. Either make a part of this study or remove result of and reference to the unpublished work.*

**REPLY**: We have clarified the text relative to these added EEMs (lines 226-228). The added samples have no link with the experiment described in the article but were needed to increase the size of the dataset used to develop the PARAFAC model. They were only used for this.

***Results***: *Please present the changes in pH if any.*

**REPLY**: Unfortunately, we did not follow changes in pH, so we have no values to present.

***Discussion***. *The authors do not discuss any possibility of phototrophic bacteria production. The BP by leucine is good for assessment only heterotrophic bacteria."*

**REPLY**: We do not discuss this potential contribution by phototrophic bacteria because we checked the presence of picophytoplankton with the cytometer and we did not find any, even at the end of the incubation under sunlight. This is specified in the revised manuscript (lines 250-252).

*L405-429 Was there any bacterial exometabolite production that could diminish overall DOC loss?"*

**REPLY**: This was not specifically measured, but it is certainly plausible that DOM was produced at the same time as it was consumed, as suggested by the rising concentrations of fluorophores (mainly observed in the dark). We added a sentence to acknowledge this (lines 563-565).

*L433-435 Rose Cory did not work with peatland lakes and rivers and thus irrelevant in this paragraph.*

**REPLY**: We removed this sentence as suggested.

*It is a bit surprising that a seminal paper on biodegradation is not cited in this manuscript. I can only guess that it is not an intentional action from the senior authors but simple negligence from their younger colleague. My advice for the young researcher would be to check the keywords for the articles in the journal before submitting a manuscript. (Vonk, J. E., Tank, S. E., Mann, P. J., Spencer, R. G. M., Treat, C. C., Striegl, R. G., Abbott, B. W., and Wickland, K. P.: Biodegradability of dissolved organic carbon in permafrost soils and aquatic systems: a meta-analysis, Biogeosciences, 12, 6915–6930, https://doi.org/10.5194/bg-12-6915-2015, 2015). In particular, Vonk et al here discusses the seasonal effects on DOC biodegradation.*

**REPLY**: Thanks for the suggestion. We have added this interesting paper in the discussion part about biodegradation (lines 511-515). However, note that this paper is mainly dealing with lotic ecosystems, which have obviously very different biogeochemical dynamics from lakes. Also, they do not have many data points in winter, but much more in spring when DOM pool in rivers is receiving strong inputs from snowmelt. By comparison, our experimental water was sampled at the early end of the winter, when there is no autotrophic production in the system and no inputs from the landscape.

*L 430-431 and L 443-444* are somehow inconsistent; may be tone down the statement in L 430-431 or be more specific about the season.

360 **REPLY**: We agree and have adjusted the text to tone down this point (lines 451-4522 and 461-462).

*L451-454 May be provide the maximal range of this process*

**REPLY**: We added the ratio provided by Moran and Zepp (1997) in lines 471-472.

*L 489-490 The reference is interesting; however, without comparison of this duration with water residence time in specific reservoirs, it does not add anything useful. It absolutely does not matter if*
365 *half of DOC is biodegraded over 2.5 years if water resides in a given pond or streams for less than a few days or weeks.*

**REPLY**: We agree, but this reference was rather added to illustrate the importance of time-scales when considering DOM degradation experiments. We added a sentence to raise the importance of considering residence time (lines 517-518).

370 *L570 Unclear what is complex pattern of sunlight exposure.*

**REPLY**: We have adjusted the text to clarify this point (line 616). We replaced the expression "complex pattern of sunlight exposure" with "complex DOM pool containing molecules photodegraded at various degrees".

*L590-591 The 7-cm depth of this study is strongly inconsistent with 0.8-m depth (L586) of other studies.*
375 *Explain he context more specifically, i.e., humic vs non-humic lakes.*

**REPLY**: The 7-cm depth given here corresponds to the experimental bottle diameter containing the lake water exposed under a thin layer of clear tap water, constraining the exposition during the incubation. We assume that in this comment, you rather refer to the discussion where we describe the attenuation depth measured in our study lake (lines 635-638); these were indeed shallower by
380 comparison to the depth indicated in the paper from Koehler et al. (2014). We adjusted the text to draw the reader's attention to the extreme humic characteristic of our study lake (lines 638-641).

**Cited references:**

Folhas, D., Duarte, A. C., Pilote, M., Vincent, W. F., Freitas, P., Vieira, G., Silva, A. M. S., Duarte, R. M. B. O. and Canário, J.: Structural characterization of dissolved organic matter in permafrost peatland lakes,
385 Water, 12(11), doi:10.3390/w12113059, 2020.

Koehler, B., Landelius, T., Weyhenmeyer, G. A., Machida, N. and Tranvik, L. J.: Sunlight-induced carbon dioxide emissions from inland waters, Global Biogeochem. Cycles, 28(7), 696–711, doi:10.1002/2014GB004850, 2014.

Matveev, A., Laurion, I. and Vincent, W. F.: Winter Accumulation of Methane and its Variable Timing
390 of Release from Thermokarst Lakes in Subarctic Peatlands, J. Geophys. Res. Biogeosciences, 124, 3521–3535, doi:10.1029/2019JG005078, 2019.

Moran, M. A. and Zepp, R. G.: Role of photoreactions in the formation of biologically labile compounds from dissolved organic matter, Limnol. Oceanogr., 42(6), 1307–1316, doi:10.4319/lo.1997.42.6.1307, 1997.

395

---

## Author Response (AR2)

Dear Helge Niemann,

We thank you for your evaluation report. We modified the manuscript to remove any remaining ambiguity raised by Reviewer 1. Please find our answers below.

Best regards,

Flora Mazoyer, Isabelle Laurion and Milla Rautio
* * *
**Answers to comments from Reviewer 1**

Dear Reviewer 1,

We thank you for taking the time to review our work a second time. Line numbers in the new version of the manuscript (version3) are specifically indicated in blue, while they are left in black when it refers to the previous version (version2).

**Shortcoming 1:** Thank you for discussing how the 4-months storage period could have impacted DOM composition and bacterial community composition during the experiments in the methods and discussion. While this is now mentioned in the main text, there is a lack of data to support the conclusions written in the manuscript.

- Lines 106-109: Without measurements of the lake water chemistry and bacterial communities at ice-off (30 May), there is no way for the authors to know whether these variables were similar in their water sample after 2 months. There are no data supporting the authors assumption that there was only a 2-month delay (instead of the actual 4-month delay) between water collection and the incubation experiments. Thus, any mention of the 2-month delay should be removed from the manuscript.

- Lines 524-525: How can the authors confirm that there was no detectable change in DOC concentration between water collection (19.2 mgC L-1) and T0 (22.1 ± 0.7 mgC L-1) if statistical tests are not performed? A 15% increase in DOC concentration is substantial and suggests microbial activity is taking place (either from the active release of exometabolites or the decay of dead microbial cells as community composition shifts). Without the statistical tests, the authors have no way to prove that the change in DOC was not statistically significant and thus need to discuss possible causes for this change.

**REPLY:** You are right that we lack data demonstrating that changes naturally happening under the ice between March (water collection) and May (ice-off) are similar to the changes observed during the storage period. We still think a 4-months delay should not make much difference with a 2-months delay and could not invalidate or bias our conclusions, but we can only assume changes happening in the refrigerated container were similar to natural changes happening under the ice. We modified the discussion to be more accurate about what is supported by our data.

Specifically, about lines 106-109: We erased all mentions of the 2-months delay to remain factual, as suggested. Therefore, we erased the sentence previously at lines 106-109, and we modified the text at lines 519 (522) and 533 (538).

Specifically, about lines 524-525: We did not say there was *no detectable change* in DOC concentration between water collection and T0, but that there was *no DOC decline*. We cannot properly test this statistically because replicate samples were not taken on the sampling day. This sentence was intended to underline the absence of any observable DOC decline over the storage period (i.e. potentially limited conversion of DOC into $CO_2$ or biomass), supporting the hypothesis that winter DOM was relatively biorecalcitrant. We are unsure what caused the DOC to increase, but agree that increasing FDOM suggests that some microbial activity was taking place (see lines 530-532). Therefore, bacterial activity may have been responsible for part of this DOC release, but it would need to come from the particulate fraction (POC). Alternatively, DOC could have leached naturally from POC, without the action of bacteria. The collected winter water was indeed containing particles and flocs visible to the naked eye, but this was not quantified. The carbon released into the dissolved fraction needs to come from somewhere, and it is very unlikely that ~2 mg $L^{-1}$ of DOC would have originated solely from already-present bacterial cells alone (as exometabolites or dead cell senescence). We thus propose that it would originate from the POC fraction, and reworked the paragraph accordingly (lines 527-530).

**Shortcoming 2:** Although the study site in Laurion et al. (2021) was much different from the lakes in this manuscript, it would still be helpful to compare their results to Laurion et al. (2021) given that there are not many measures of DOM processing in wetland thermokarst lakes.

**REPLY:** After rethinking the question, we agree with you because these two sites have enough in common to bring a meaningful comparison, which we added in the photodegradation part of the discussion (lines 457-458) and in the biodegradation parts of the discussion (lines 546-548 and 581-582).
* * *
**Answers to comments from Reviewer 2**

Dear Reviewer 2,

We thank you for your time reviewing our work. Your suggestions of language improvements were very much appreciated.